# ROBUST BACKDOOR REMOVAL BY RECONSTRUCTING TRIGGER-ACTIVATED CHANGES IN LATENT REPRESENTATION

## ABSTRACT

Backdoor attacks pose a critical threat to machine learning models, causing them to behave normally on clean data but misclassify poisoned data into a poisoned class. Existing defenses often attempt to identify and remove backdoor neurons based on Trigger-Activated Changes (TAC) which is the activation differences between clean and poisoned data. These methods suffer from low precision in identifying true backdoor neurons due to inaccurate estimation of TAC values. In this work, we propose a novel backdoor removal method by accurately reconstructing TAC values in the latent representation. Specifically, we formulate the minimal perturbation that forces clean data to be classified into a specific class as a convex quadratic optimization problem, whose optimal solution serves as a surrogate for TAC. We then identify the poisoned class by statistical test based on extreme selection bias of the class with the smallest norm of perturbations, and leverage the perturbation of the poisoned class in fine-tuning to remove backdoors. Experiments on CIFAR-10, GTSRB, and TinyImageNet demonstrated that our approach consistently achieves superior backdoor suppression with high clean accuracy across different attack types, datasets, and architectures, outperforming existing defense methods.

## 1 INTRODUCTION

While machine learning provides significant benefits in many applications, the threat of backdoor attacks that compromise machine learning models has been pointed out (Gu et al., 2019; Chen et al., 2017; Nguyen & Tran, 2021). The compromised model behaves normally on clean data, but when a trigger known only to the adversary is embedded into the data (poisoned data), the model is forced to misclassify it as the attacker-specified target class. One of the most critical challenges in backdoor defenses is to develop backdoor removal methods that effectively eliminate the influence of backdoor attacks from a compromised model while preserving its original accuracy (Liu et al., 2018a; Zheng et al., 2022; Lin et al., 2024).

To minimize accuracy degradation, most backdoor removal methods first identify backdoor neurons that strongly respond to the trigger and are thus thought to be less essential for normal predictions but critical for backdoor success. Once identified, the influence (impact) of these neurons is mitigated through pruning, fine-tuning or both (Liu et al., 2018a; Zheng et al., 2022; Wu & Wang, 2021; Li et al., 2023; Lin et al., 2024). A key metric to measure the degree of their contribution is Trigger-Activated Changes (TAC) (Zheng et al., 2022), defined as the difference in neuron activations between clean and poisoned data. Removing neurons exhibiting higher TAC values can eliminate backdoors while minimizing the impact on accuracy (Zheng et al., 2022; Lin et al., 2024).

However, since poisoned data is not available in practice, the ideal values of TAC cannot be obtained. Due to this limitation, existing methods (Liu et al., 2018a; Zheng et al., 2022; Wu & Wang, 2021; Li et al., 2023; Lin et al., 2024) compute the contribution of neurons to the success of backdoor attacks using their own approaches, but their results often show low consistency with TAC, leading to ineffective backdoor removal.

To address this problem, we propose a novel backdoor removal method by accurately reconstructing the effects of TAC in the latent representation with an overview provided in Figure 1. Among

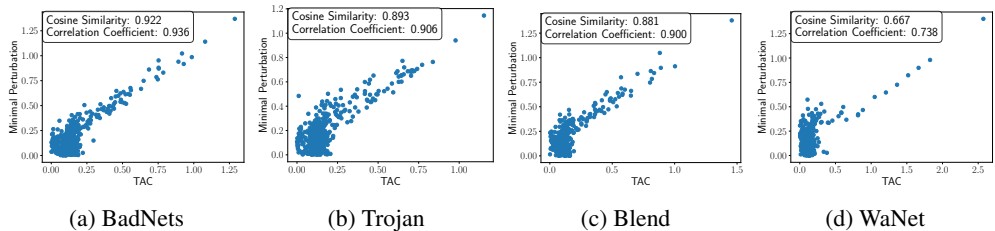

Figure 1: Overview of our proposed method. Our method consists of two stages: (1) reconstructing TAC in the latent representation, which involves computing the minimal perturbation that forces any clean data to be classified into each class and then identifying the poisoned class with statistical test via extreme selection bias for the class with the smallest norm, and (2) removing the backdoor by fine-tuning with the optimized perturbation of the poisoned class.

| (a) BadNets | (b) Trojan | (c) Blend | (d) WaNet |

Figure 2: The perturbations obtained by our method and TAC in the latent representation for CIFAR-10 on ResNet-18. For each neuron in the latent representation, we plot the TAC value on the horizontal axis and the minimal perturbation of the poisoned class on the vertical axis.

intermediate layers, TAC in the latent representation, i.e, the output of the layer just before the classification layer, can be critical for the success of backdoor attacks because the effects of TAC in earlier layers propagate and accumulate in the latent representation, which then directly affects misclassification through the classification layer. If the effects of TAC in the latent representation can be inferred solely from clean data, defenders can approximate the model's outputs on poisoned data without them and eliminate their influence from the model. Thus, reconstructing TAC in the latent representation enables robust backdoor removal.

Specifically, we first reconstruct TAC in the latent representation by computing a minimal perturbation in that representation required to misclassify any clean data into the poisoned class. This is motivated by two key properties of TAC in the latent representation: (i) because triggers are realized through minimal modifications to clean data in order to remain undetectable, the resulting changes (i.e., TAC) in the latent representation between clean and poisoned inputs are necessarily small; and (ii) despite being minimal, these changes are sufficient to induce misclassification into the poisoned class. Actually, Figure 2 shows that the minimal perturbation obtained in this way is strongly similar and correlated with TAC in the latent representation. We then apply the obtained perturbation for model fine-tuning, which effectively removes the backdoor while preserving clean accuracy.

Our main contributions are summarized as follows:

**1. Method to Reconstruct TAC in the Latent Representation.** We propose a method to reconstruct TAC in the latent representation by computing the minimal perturbation that forces any clean data to be misclassified into a specific class and identifying the poisoned class from the perturbations in all classes. First, we formulate the optimization problem of finding such a perturbation as a convex quadratic program. We then clarify the conditions under which such a perturbation exists and derive the analytical solution. For the poisoned class, the perturbation obtained by solving the optimization can be regarded as a surrogate that reproduces the effect of TAC. Therefore, reconstructing

TAC in the latent representation requires identifying the poisoned class, even though in practice the defender typically does not know it in advance.

**2. Statistical Identification of the Poisoned Class via Extreme Selection Bias.** We propose a statistical method for identifying the poisoned class based on the frequency of the class with the smallest norm among the perturbations of all class. Backdoor training forces data with triggers to be classified into the poisoned class by effectively shifting its decision boundary toward the region of clean data so the perturbation norm of the poisoned class is smaller than that of other classes. Due to this property, the poisoned class tends to be extremely selected as the class with the smallest norm, whereas this does not occur in clean models. By formulating this phenomenon as a statistical test, we obtain a statistically reliable method.

**3. Backdoor Removal Method from the Optimized Perturbation.** We propose a backdoor removal method that leverages the TAC effects estimated by our method of the poisoned class. Concretely, we fine-tune the model using a loss that enforces clean data in the latent representation, even when perturbed toward the poisoned class, to be classified into their original clean classes, together with the cross-entropy loss for the clean task. This process yields a compromised model that simultaneously preserves high accuracy and enhances backdoor removal performance. Experimental results demonstrate that our method can successfully eliminate the impact of backdoor attacks while maintaining high accuracy, even against several representative attack methods. Furthermore, we confirm that our approach achieves greater robustness compared to existing defense methods.

## 2 RELATED WORKS

### 2.1 BACKDOOR ATTACKS

A backdoor attack compromises a model so that it behaves normally on clean data but misclassifies poisoned data into an attacker-specified class. Representative methods include BadNets (Gu et al., 2019), Blend (Chen et al., 2017), and Trojan (Liu et al., 2018b). Although these approaches achieve high attack success rates, they are relatively easy to detect because of their easily visible triggers. To reduce the detectability of visible triggers, several studies design imperceptible triggers such that the difference between clean and poisoned data cannot be distinguished by humans or detectors (Nguyen & Tran, 2020; 2021; Doan et al., 2021b). More recently, techniques have also been developed to improve stealthiness not only at the input level but also in the internal feature space of the model (Tan & Shokri, 2020; Zhong et al., 2022; Doan et al., 2021a; Xu et al., 2025). In this way, backdoor attacks continue to evolve toward greater stealthiness in both input and internal space, thereby increasing the difficulty of effective defense.

### 2.2 BACKDOOR REMOVAL

Existing backdoor removal methods can broadly be categorized into two groups: (i) those that identify backdoor neurons and then prune or fine-tune them (Liu et al., 2018a; Zheng et al., 2022; Wu & Wang, 2021; Li et al., 2023; Lin et al., 2024), and (ii) those that neutralize backdoors via advanced fine-tuning strategies without explicit neuron identification (Zhu et al., 2023; Min et al., 2023; Wei et al., 2023; Karim et al., 2024). Details of the latter related works are provided in Appendix B.1.

To identify backdoor neurons, various methods have been proposed. Fine-Pruning (FP) (Liu et al., 2018a) regards neurons inactive on clean data as backdoor neurons, while Adversarial Neuron Pruning (ANP) (Wu & Wang, 2021) regards neurons sensitive to adversarial noise as backdoor neurons. As an oracle metric, Channel Lipschitzness Pruning (CLP) (Zheng et al., 2022) introduced Trigger-Activated Changes (TAC), defined as the activation difference between clean and poisoned data. CLP further approximates neurons with large weight values as those with large TAC. However, because TAC computation requires access to poisoned data, it is impossible to obtain the ideal values of TAC. More recently, unlearning-based methods using only clean data (Li et al., 2023; Lin et al., 2024) have been proposed, but the identification rate of neurons with high TAC values still remains limited. If TAC could be computed more precisely, it would enable approximate the model outputs of poisoned data without them and thus achieve robust backdoor removal. However, in the absence of poisoned data, directly leveraging TAC is infeasible, leaving the construction of practical surrogates of TAC for defenders as an open challenge. To address this challenge, our approach reconstructs TAC in the latent representation via optimizing a minimal perturbation that forces any clean data

to be misclassified into a specific class, providing a feasible and accurate method for defenders to neutralize backdoor effects.

# 3 PROBLEM SETTING

In this section, we first describe the threat model in this work, focusing on the goals and capabilities of the adversary and the defender. We then present the formalization of backdoor attacks and introduce Trigger-Activated Changes (TAC) (Zheng et al., 2022).

## 3.1 THREAT MODEL

**Adversary**. The adversary's goal is to obtain a compromised model that, with high probability, misclassifies poisoned data into the target class while still correctly classifying clean data. In the data collection scenario (Gu et al., 2019; Chen et al., 2017; Liu et al., 2020), the adversary has access only to the training dataset. In the supply-chain scenario (Doan et al., 2021b; Nguyen & Tran, 2021; Xu et al., 2025), where the model is distributed through external sources, the adversary may have full access to the training process.

**Defender.** The defender's goal is to detect whether a given model has been compromised and to remove the backdoor if present. The defender is assumed to have access to the model parameter and a small dataset (reference dataset) sampled from the same distribution as the model's training data (Zhu et al., 2023; Lin et al., 2024).

## 3.2 FORMULATION OF BACKDOOR ATTACKS

For any $a \in \mathbb{N}$, let $[a] = \{1, 2, \cdots, a\}$. Given an input dimension $d_{\text{in}}$ and the number of classes $C$, we denote by $e_i \in \{0, 1\}^C$ the standard basis vector whose $i$-th element is 1. A neural network $f : \mathbb{R}^{d_{\text{in}}} \to [0, 1]^C$ outputs the probability of belonging to each class for an input $\boldsymbol{x} \in \mathbb{R}^{d_{\text{in}}}$. Let $\boldsymbol{\theta}$ be a model parameter, $\ell$ a loss function, and $\phi_{\boldsymbol{\theta}} : \mathbb{R}^{d_{\text{in}}} \to \mathbb{R}^{d_{\text{emb}}}$ the mapping to the latent representation layer (i.e., the layer just before the final linear layer) of dimension $d_{\text{emb}}$. This yields the latent representation $\hat{\boldsymbol{x}} = \phi_{\boldsymbol{\theta}}(\boldsymbol{x}) \in \mathbb{R}^{d_{\text{emb}}}$. The final ($L$-th) linear layer is parameterized by weight matrix $\boldsymbol{W}_L = [\boldsymbol{w}_1, \boldsymbol{w}_2, \cdots, \boldsymbol{w}_C] \in \mathbb{R}^{d_{\text{emb}} \times C}$, where each column vector is $\boldsymbol{w}_j \in \mathbb{R}^{d_{\text{emb}}}$ and a bias vector is $\boldsymbol{b} \in \mathbb{R}^C$. Using the softmax function, the network output is expressed as $f(\boldsymbol{x}; \boldsymbol{\theta}) = \text{Softmax}(\boldsymbol{W}_L^\top \hat{\boldsymbol{x}} + \boldsymbol{b})$.

Furthermore, let $\boldsymbol{\delta} \in \mathbb{R}^{d_{\text{in}}}$ be the trigger required for a backdoor attack and $t \in [C]$ be a poisoned class. Then, the compromised model parameter $\boldsymbol{\theta}_{\text{bd}}$ is obtained as

$$\boldsymbol{\theta}_{\text{bd}} = \operatorname*{argmin}_{\boldsymbol{\theta}} \frac{1}{n} \sum_{i=1}^{n} \left[ \ell(f(\boldsymbol{x}_i; \boldsymbol{\theta}), \boldsymbol{y}_i) + \ell(f(\boldsymbol{x}_i + \boldsymbol{\delta}; \boldsymbol{\theta}), \boldsymbol{e}_t) \right], \tag{1}$$

where the training dataset is $D = \{(\boldsymbol{x}_i, \boldsymbol{y}_i)\}_{i=1}^{n}$ and $\boldsymbol{y}_i \in \{\boldsymbol{e}_1, \boldsymbol{e}_2, \cdots, \boldsymbol{e}_C\}$. The parameter $\boldsymbol{\theta}_{\text{bd}}$ is optimized such that the model behaves normally on clean data $\boldsymbol{x}$, while poisoned data $\boldsymbol{x} + \boldsymbol{\delta}$ are misclassified into the poisoned class $t$.

## 3.3 TRIGGER-ACTIVATED CHANGES

In a compromised model, when poisoned data $\boldsymbol{x} + \boldsymbol{\delta}$ is provided, certain neurons are strongly activated. This excessive activation causes $\boldsymbol{x} + \boldsymbol{\delta}$ to be misclassified into the poisoned class $t$. Therefore, if the contribution of each neuron to the success of the backdoor attack can be quantified, its influence can be suppressed, enabling backdoor removal from the model.

In this paper, we focus on Trigger-Activated Changes (TAC) (Zheng et al., 2022), which are defined as the difference in activations between clean and poisoned data and serve as an oracle metric to quantify each neuron's contribution to the success of backdoor attacks. Specifically, for the $i$-th neuron in the $l$-th layer $f_{l,i}(\cdot)$, TAC is computed as

$$\text{TAC}_{l,i}(\boldsymbol{x}; \boldsymbol{\theta}) = f_{l,i}(\boldsymbol{x} + \boldsymbol{\delta}; \boldsymbol{\theta}) - f_{l,i}(\boldsymbol{x}; \boldsymbol{\theta}). \tag{2}$$

The $i$-th neuron's importance in the intermediate layers for the success to backdoor attack is calculated as the average value, $\mathrm{TAC}_{l,i}(\boldsymbol{\theta}) = \mathbb{E}_{\boldsymbol{x}}[\mathrm{TAC}_{l,i}(\boldsymbol{x};\boldsymbol{\theta})]$, because applying the same trigger to different data tends to activate similar neurons in the intermediate layers (Zheng et al., 2022).

However, the computation of TAC requires poisoned data $\boldsymbol{x} + \boldsymbol{\delta}$, and since defenders typically do not know the trigger $\boldsymbol{\delta}$, it is infeasible to calculate the ideal values of $\mathrm{TAC}_{l,i}(\boldsymbol{\theta})$.

# 4 PROPOSED METHOD

In this paper, we aim to reconstruct TAC in the latent representation of a compromised model instead of reconstructing TAC in arbitrary intermediate layers. The rationale is that although backdoor neurons may appear in arbitrary intermediate layers, their effects are aggregated through the network and ultimately reflected in the latent representation. Thus, if TAC in the latent representation can be reconstructed, the output of poisoned data can be approximately computed using the subsequent linear layer as follows: $f(\boldsymbol{x} + \boldsymbol{\delta};\boldsymbol{\theta}) \coloneqq \mathrm{Softmax}(\boldsymbol{W}_L(\hat{\boldsymbol{x}} + \mathrm{TAC}_{L-1}(\boldsymbol{\theta})) + \boldsymbol{b})$, where $\mathrm{TAC}_{L-1}(\boldsymbol{\theta}) \in \mathbb{R}^{d_{\mathrm{emb}}}$ denotes the vector of TAC values in the latent representation. This allows us to remove backdoors by fine-tuning the model so that the misclassification of poisoned data is restored into the correct class.

Based on this idea, we propose a method to reconstruct TAC in the latent representation and a defense mechanism that leverages the reconstructed TAC for backdoor removal. As illustrated in Figure 1, our method consists of two stages: (1) reconstructing TAC in the latent representation, which involves computing the minimal perturbation that forces any clean data to be classified into each class and then identifying the poisoned class with statistical test via extreme selection bias for the class with the smallest norm, and (2) removing the backdoor using the optimized perturbation of the poisoned class. The details of each stage are described below.

## 4.1 COMPUTING PERTURBATIONS IN THE LATENT REPRESENTATION

To reconstruct TAC in the latent representation, we focus on the following two properties of TAC in the latent representation: it takes minimal values since the trigger is minimized to be indistinguishable from the original data, and it induces misclassification into the poisoned class. Based on these observations, we first introduce an optimization problem to compute the minimal perturbation in the latent representation of clean data that forces it to be misclassified into a specific class.

**Optimization Problem.** Our goal is to find the minimal perturbation $\boldsymbol{s}_k^*$ that guarantees all inputs are classified into class $k$. This leads to the following formulation: the objective is defined by a quadratic term $\frac{1}{2}\|\boldsymbol{s}_k\|_2^2$ for analytical convenience, such that the logits $\boldsymbol{s}_k + \hat{\boldsymbol{x}}_i$ of class $k$ dominate those of all other classes. The resulting primal optimization problem can be formulated as the following convex quadratic program:

$$\boldsymbol{s}_k^* = \operatorname*{argmin}_{\boldsymbol{s}_k} \ \frac{1}{2}\|\boldsymbol{s}_k\|_2^2 \quad \text{s.t.} \quad (\boldsymbol{w}_k - \boldsymbol{w}_j)^\top(\boldsymbol{s}_k + \hat{\boldsymbol{x}}_i) \ \geq \ 0, \ \ \forall j \in [C] \setminus \{k\}, \ \forall i \in [n]. \quad (3)$$

Here, $(\boldsymbol{w}_k - \boldsymbol{w}_j)^\top(\boldsymbol{s}_k + \hat{\boldsymbol{x}}_i)$ denotes the margin of class $k$ against class $j$ for sample $i$ after applying the perturbation $\boldsymbol{s}_k$. An example $\boldsymbol{x}_i$ is classified into class $k$ if and only if these margins are nonnegative for all $j \neq k$. Therefore, the constraints enforce nonnegative margins for every example and every $j \neq k$, and the single perturbation $\boldsymbol{s}_k$ is chosen to lift the margins of class $k$ simultaneously across all examples. To remove redundancy and improve computational efficiency, the $n(C-1)$ constraints in equation 3 are compressed into $C-1$ constraints by considering only the worst-case margin for each class $j \neq k$ across the dataset. That is, the constraint in equation 3 can be equivalently written as $(\boldsymbol{w}_k - \boldsymbol{w}_j)^\top \boldsymbol{s}_k \geq -(\boldsymbol{w}_k - \boldsymbol{w}_j)^\top \hat{\boldsymbol{x}}_i, \quad \forall j \in [C] \setminus \{k\}, \quad \forall i \in [n]$ and it suffices to consider only the worst case $\forall j \in [C] \setminus \{k\} : \max_i \{-(\boldsymbol{w}_k - \boldsymbol{w}_j)^\top \hat{\boldsymbol{x}}_i\}$. The problem therefore reduces to the following convex quadratic program:

$$\boldsymbol{s}_k^* = \operatorname*{argmin}_{\boldsymbol{s}_k} \frac{1}{2}\|\boldsymbol{s}_k\|_2^2 \quad \text{s.t.} \quad \boldsymbol{U}_k \boldsymbol{W}_L^\top \boldsymbol{s}_k \geq \boldsymbol{m}, \quad (4)$$

where the inequality between vectors is understood element-wise, $\boldsymbol{U}_k \coloneqq [\boldsymbol{u}_1, \boldsymbol{u}_2, \cdots, \boldsymbol{u}_{C-1}] \in \mathbb{R}^{(C-1)\times C}, \forall j \in [C] \setminus \{k\} : \boldsymbol{u}_j = (\boldsymbol{e}_k - \boldsymbol{e}_j)^\top \in \mathbb{R}^C$ and $\boldsymbol{m} \in \mathbb{R}^{C-1}$ is the vector of worst-case margins, with each component given by $\forall j \in [C] \setminus \{k\} : \boldsymbol{m}_j = \max_i \{-(\boldsymbol{w}_k - \boldsymbol{w}_j)^\top \hat{\boldsymbol{x}}_i\}$.

The reduced problem is also convex by construction but its feasibility is not always guaranteed. Thus, we provide sufficient conditions under which feasibility is guaranteed from Theorem 1. That is, if $C - 1 < d_{\text{emb}}$ and $\boldsymbol{U}_k \boldsymbol{W}_L^\top$ has full row rank, the optimal solution $\boldsymbol{s}_k^*$ is guaranteed to exist.

**Solution via Dual Problem**. To obtain the optimal solution for $\boldsymbol{s}_k$, we introduce the dual problem of equation 4 because the dual problem involves fewer variables, which makes the problem more stable compared to the primal problem. Let $\boldsymbol{\lambda} \in \mathbb{R}^{C-1}$ be the dual variable vector and $\boldsymbol{V}_k := \boldsymbol{U}_k \boldsymbol{W}_L^\top \in \mathbb{R}^{(C-1) \times d_{\text{emb}}}$. The final form of the dual problem can be written as follows, with the derivation process provided in Appendix D.2:

$$\boldsymbol{\lambda}^* = \underset{\boldsymbol{\lambda}}{\operatorname{argmax}} \quad \boldsymbol{\lambda}^\top \boldsymbol{m} - \frac{1}{2} \| \boldsymbol{V}_k^\top \boldsymbol{\lambda} \|_2^2 \quad \text{s.t.} \quad \boldsymbol{\lambda} \geq \boldsymbol{0}. \tag{5}$$

In general, the dual problem provides a lower bound on the optimal value of the primal problem. When the primal problem is convex and satisfies suitable regularity conditions (e.g., Slater's condition (Boyd & Vandenberghe, 2004)), strong duality holds, and the optimal values of the primal and dual problems coincide. The proof that strong duality for the derived primal and dual problems in equation 4 and equation 5 is given in Appendix D.4. When strong duality holds, the following Karush–Kuhn–Tucker (KKT) conditions are necessary and sufficient for optimality of the primal problem (Boyd & Vandenberghe, 2004):

(i) Stationarity: $\boldsymbol{s}_k^* - \boldsymbol{V}^\top \boldsymbol{\lambda}^* = \boldsymbol{0}$,

(ii) Primal and Dual Feasibility: $\boldsymbol{V}^\top \boldsymbol{s}_k^* \geq \boldsymbol{m}, \boldsymbol{\lambda}^* \geq \boldsymbol{0}$,

(iii) Complementary Slackness: $\boldsymbol{\lambda}^* \odot (\boldsymbol{V}^\top \boldsymbol{s}_k^* - \boldsymbol{m}) = \boldsymbol{0}$.

The conditions of primal and dual feasibility together with complementary slackness ensure that $\boldsymbol{s}_k^*$ necessarily induces misclassification into class $k$. As a result, the minimal perturbation $\boldsymbol{s}_k^*$ can be obtained from the stationarity condition, i.e., $\boldsymbol{s}_k^* = \boldsymbol{V}^\top \boldsymbol{\lambda}^*$, which shows that the primal optimal solution can be obtained from the dual optimal solution. In practice, we solve the dual problem with a convex optimization solver CVXPY (Diamond & Boyd, 2016) to obtain $\boldsymbol{\lambda}^*$ reliably.

## 4.2 IDENTIFYING THE POISONED CLASS VIA EXTREME SELECTION BIAS

Using the perturbations for each class obtained in Section 4.1, we propose a method to identify the poisoned class. Since TAC in the latent representation is reconstructed as the perturbation $\boldsymbol{s}_t^*$ for the poisoned class $t \in [C]$, it is necessary to identify the poisoned class.

To this end, we first focus on the perspective of $L^2$ minimization of the perturbations. The minimal displacement required to switch class is proportional to the margin to the decision boundary. Backdoor training tends to pull the poisoned decision boundary closer to the data space with clean classes, which causes the perturbation of the poisoned class to become smaller than those of the other classes. Additionally, we leverage a key empirical property: in compromised models, the poisoned class almost always appears as the class with the smallest perturbation norm $\| \boldsymbol{s}_k^* \|_2$, even when the computation of $\boldsymbol{s}_1^*, \boldsymbol{s}_2^*, \cdots, \boldsymbol{s}_C^*$ is repeated many times. In contrast, clean models may show small selection bias across classes, but they never exhibit such extreme selection bias on a single class such as compromised models.

Concretely, we first calculate the latent representations $\hat{\boldsymbol{x}}_1, \hat{\boldsymbol{x}}_2, \cdots, \hat{\boldsymbol{x}}_n$ of the entire reference dataset which a defender has in advance. Then, we randomly sample a subset of them at a sampling rate $r$, and obtain $\boldsymbol{s}_1^*, \boldsymbol{s}_2^*, \cdots, \boldsymbol{s}_C^*$ by solving the optimized problem as described in 4.1. This process is repeated $T_{\text{pci}}$ times and we record which class attains the minimum norm in each process. Let $N_k$ be the number of times class $k$ is selected, $N^* = \max_{1 \leq k \leq C} N_k$, and $p_{\max} = N^* / T_{\text{pci}}$. We then formulate a hypothesis test on this maximum selection ratio. A clean model is defined as one whose maximum selection probability does not exceed a tolerance level $\eta$; a compromised model violates this bound. Formally, we consider the one-sided test

$$H_0 : p_{\max} \leq \eta \quad \text{vs} \quad H_1 : p_{\max} > \eta. \tag{6}$$

Since $H_0$ is a composite hypothesis, we evaluate the $p$-value under the least favorable clean case, where one class has selection probability exactly $\eta$ and the others are ignored. In this case, $N^*$ is

stochastically dominated by a binomial random variable $X \sim \text{Binomial}(T_{\text{pci}}, \eta)$, and we define the $p$-value as $p_{\text{val}} = \Pr[X \geq N^*]$. In other words, we deliberately consider the clean model in which a large value of $N^*$ is most likely to occur, and ask how plausible the observed value is under the clean model. In this way, the $p$-value quantifies how natural it would be, under $H_0$, for a clean model to select the same class as the smallest norm class as many times as we observed. We then declare the model compromised if $p_{\text{val}} < \alpha$, and identify the poisoned class as the one with the largest $N_k$. Otherwise, we regard the observed bias as consistent with a clean model. After identifying the poisoned class, we compute the minimal perturbation $\boldsymbol{s}_t^*$ using the entire reference dataset.

### 4.3 BACKDOOR REMOVAL WITH THE PERTURBATION OF THE POISONED CLASS

Using the perturbation of the poisoned class obtained in Section 4.2, we propose a backdoor removal method, as shown in equation 7:

$$\theta_{\text{ft}}^* = \underset{\boldsymbol{\theta}_{\text{bd}}}{\arg\min} \frac{1}{n} \sum_{i=1}^{n} \left[ \ell(f(\boldsymbol{x}_i; \boldsymbol{\theta}_{\text{bd}}), \boldsymbol{y}_i) + \beta \, \ell\big(\text{Softmax}(\boldsymbol{W}_L^\top(\hat{\boldsymbol{x}}_i + \boldsymbol{s}_t^*) + \boldsymbol{b}), \boldsymbol{y}_i\big) \right], \tag{7}$$

where $\beta$ is a hyperparameter that balances clean accuracy and backdoor removal performance. Specifically, the model is fine-tuned so that even if the latent representation shifts in the direction of $\boldsymbol{s}_t^*$, the perturbed latent representation $\hat{\boldsymbol{x}}_i + \boldsymbol{s}_t^*$ is still recognized as its correct class. This ensures that poisoned data are classified into their correct clean classes. In addition, to maintain performance on the original clean task, a loss that enforces correct classification of clean data into their correct classes is also included.

While pruning-based approaches via the perturbation of the poisoned class are also possible, we found that our fine-tuning method is more effective performance as shown in Appendix F.4.

## 5 EXPERIMENTS

In this section, we conduct experimental evaluations to verify the effectiveness of our proposed method described in Section 4.

### 5.1 EXPERIMENTAL SETUP

**Datasets and Neural Network Architecture.** We conducted experiments on three image classification datasets: CIFAR-10, GTSRB, and TinyImageNet. CIFAR-10 and GTSRB contain 10 and 43 classes of $32 \times 32$ pixels, respectively. TinyImageNet includes 200 classes with images resized to $64 \times 64$ pixels. For all datasets, we primarily used ResNet-18 as the neural network architecture.

**Backdoor Attacks.** We evaluate the effectiveness of our proposed method against six backdoor attack methods: BadNets (Gu et al., 2019), Trojan (Liu et al., 2018b), Blend (Chen et al., 2017), IAB (Nguyen & Tran, 2020), Lira (Doan et al., 2021b), and WaNet (Nguyen & Tran, 2021). The training configuration for all attacks consisted of 100 epochs, stochastic gradient descent (SGD) as the optimizer, a learning rate of 0.1, and cosine annealing as the learning rate scheduler. For the standard backdoor attack configuration, we adopted a poisoning rate of 10.0% and fixed the poisoned class as 1, following the all-to-one setting in which poisoned data from all other classes is misclassified into a poisoned class. We remark that although datasets such as CIFAR-10 index classes starting from 0, we align with the notation in this paper where classes are indexed from 1. Accordingly, class 1 in our notation corresponds to class 0 in CIFAR-10. Further details of each attack and the hyperparameters used are provided in Appendix C.1.

**Backdoor Defenses.** For comparison, we evaluate our proposed method against five defense methods to identify backdoor neurons, FP (Liu et al., 2018a), CLP (Zheng et al., 2022), ANP (Wu & Wang, 2021), RNP (Li et al., 2023) and TSBD (Lin et al., 2024) as well as three advanced fine-tuning defenses without identifying backdoor neurons, FT-SAM (Zhu et al., 2023), SAU (Wei et al., 2023) and FST (Min et al., 2023). Details of the defense methods and hyperparameters are provided in Appendix C.2. Following previous works (Zhu et al., 2023; Lin et al., 2024), we assume that the defender has access to 5% of the training dataset as a reference dataset and the effect of the reference dataset size on defense performance is presented in Appendix F.5. For our poisoned class identification method, the hyperparameters are $\alpha = 0.01$, $\eta = 0.7$, $r = 0.4$ and $T_{\text{pci}}$ for all datasets.

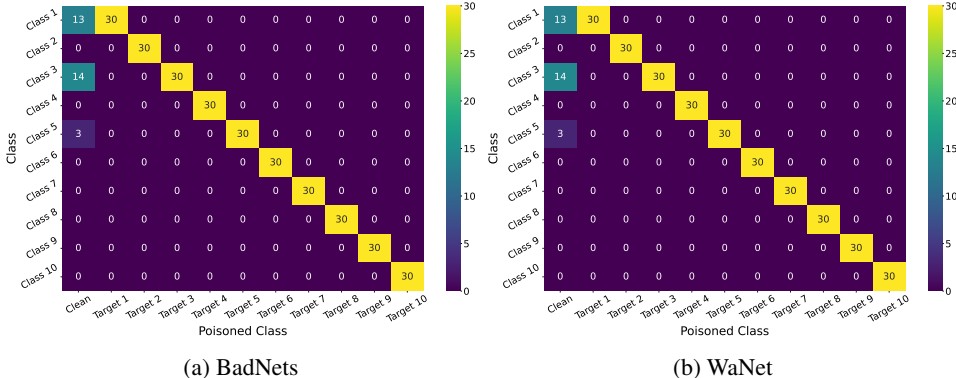

(a) BadNets                                    (b) WaNet

Figure 3: Total number of times the class with the smallest norm is counted for each poisoned class on CIFAR-10. The horizontal axis shows the poisoned class, while the vertical axis shows each class.

For our fine-tuning method, we used SGD with a learning rate of 0.01 for 50 epochs, with $\beta$ set to 0.5 for CIFAR-10, 2.0 for GTSRB, and 0.1 for TinyImageNet. The effect of tuning $\beta$ value on ACC and ASR is discussed in Appendix F.6.

## 5.2 RECONSTRUCTING TAC IN THE LATENT REPRESENTATION

As shown in Figure 2 and Appendix F.2, the perturbation of the poisoned class computed by our method exhibits a high similarity with TAC in the latent representation and identifies backdoor neurons more accurately than existing approaches. Therefore, it is crucial to accurately identify the poisoned class by our poisoned class identification method described in Section 4.2.

Table 1: Results for our poisoned class identification method. "Clean" shows the result of the clean model without any attack.

| | CIFAR-10 | | | GTSRB | | | TinyImageNet | | |
|---|---|---|---|---|---|---|---|---|---|
| | $p_{\text{val}}$ | $N^*$ | Poisoned Class | $p_{\text{val}}$ | $N^*$ | Poisoned Class | $p_{\text{val}}$ | $N^*$ | Poisoned Class |
| Clean | 0.998 | 14 | - | 0.589 | 21 | - | 0.841 | 19 | - |
| BadNets | 0.0000226 | 30 | 1 | 0.0000226 | 30 | 1 | 0.0000226 | 30 | 1 |
| Trojan | 0.0000226 | 30 | 1 | 0.0000226 | 30 | 1 | 0.0000226 | 30 | 1 |
| Blend | 0.0000226 | 30 | 1 | 0.0000226 | 30 | 1 | 0.0000226 | 30 | 1 |
| WaNet | 0.0000226 | 30 | 1 | 0.0000226 | 30 | 1 | 0.0000226 | 30 | 1 |
| IAB | 0.0000226 | 30 | 1 | 0.0000226 | 30 | 1 | 0.0000226 | 30 | 1 |
| Lira | 0.0000226 | 30 | 1 | 0.0000226 | 30 | 1 | 0.0000226 | 30 | 1 |

**Main Results.** Table 1 shows the clean model exhibits only $N^* = 20$ for CIFAR-10, resulting in an extremely large $p_{\text{val}}$ and thus is not rejected under $H_0$. In contrast, differing trigger types and injection mechanisms show perfect concentration ($N^* = 30$) in all datasets, yielding highly significant $p$-values ($< 0.001$). These results confirm that the proposed method consistently identifies the poisoned class and clearly separates compromised models from clean ones.

**Different Poisoned Classes.** To examine the robustness of our poisoned class identification method, we further evaluate whether our proposed method can detect any poisoned class as shown in Figure 3. As a result, we confirmed that regardless of which class was poisoned, the poisoned class was selected as the one with the smallest norm across all trials.

## 5.3 EFFECTIVENESS OF BACKDOOR REMOVAL

**Evaluation Metrics.** Following previous works (Zhu et al., 2023; Lin et al., 2024), we introduce three evaluation metrics for backdoor attacks. Accuracy (ACC), which measures the classification accuracy on clean data; Attack Success Rate (ASR), which denotes the percentage of triggered inputs classified into the poisoned class; and Defense Efficacy Rate (DER), which evaluates how effectively the backdoor is removed while maintaining accuracy. DER is defined as DER $= (\max(0, \Delta\text{ACC}) -$

Table 2: Comparison of the backdoor removal results. "Average" denotes the mean of each metric across attack methods. "No Defense" refers to a model to which no defense method is applied so DER is marked as "−".

**CIFAR-10**

| | | No Defense | | | FP | | | CLP | | | ANP | | | RNP | | |
|---|---|---|---|---|---|---|---|---|---|---|---|---|---|---|---|---|
| | | ACC↑ | ASR↓ | DER↑ | ACC↑ | ASR↓ | DER↑ | ACC↑ | ASR↓ | DER↑ | ACC↑ | ASR↓ | DER↑ | ACC↑ | ASR↓ | DER↑ |
| | BadNets | 93.81 | 100.00 | – | **93.68** | 100.00 | 49.94 | 91.30 | 33.10 | 82.19 | 87.89 | 2.67 | 95.46 | 93.17 | 6.90 | 95.98 |
| | Trojan | 94.00 | 100.00 | – | 93.47 | 2.11 | **98.68** | 83.97 | 1.19 | 94.39 | 88.98 | 100.00 | 47.35 | **93.97** | 99.93 | 49.86 |
| | Blend | 93.29 | 99.91 | – | 93.11 | 13.96 | 92.89 | 90.09 | 30.46 | 83.13 | 89.72 | 91.12 | 52.14 | **93.43** | 89.07 | 55.03 |
| | WaNet | 93.41 | 99.59 | – | 93.35 | 0.79 | **99.37** | 10.23 | 100.00 | 8.41 | 92.73 | 1.48 | 98.47 | **93.40** | 87.78 | 55.66 |
| | IAB | 93.57 | 98.81 | – | 93.40 | **0.32** | **99.16** | 90.18 | 7.39 | 94.02 | 89.13 | 0.83 | 96.99 | **93.57** | 1.27 | 99.00 |
| | Lira | 94.29 | 99.98 | – | **93.88** | 0.24 | **99.66** | 88.61 | 3.86 | 95.22 | 90.82 | 99.93 | 48.29 | 93.82 | 90.98 | 54.27 |
| | Average | 93.73 | 99.71 | – | 93.48 | 19.57 | 89.95 | 75.73 | 29.33 | 76.23 | 89.88 | 49.34 | 73.12 | **93.56** | 62.65 | 68.30 |

| | | TSBD | | | FT-SAM | | | SAU | | | FST | | | Ours | | |
|---|---|---|---|---|---|---|---|---|---|---|---|---|---|---|---|---|
| | | ACC↑ | ASR↓ | DER↑ | ACC↑ | ASR↓ | DER↑ | ACC↑ | ASR↓ | DER↑ | ACC↑ | ASR↓ | DER↑ | ACC↑ | ASR↓ | DER↑ |
| CIFAR-10 | BadNets | 29.22 | 91.38 | 22.02 | 92.85 | 2.78 | **98.13** | 85.84 | **1.29** | 95.37 | 93.53 | 100.00 | 49.86 | 92.03 | 10.88 | 93.67 |
| | Trojan | 90.12 | 3.57 | 96.28 | 92.88 | 2.06 | 98.41 | 90.65 | 1.71 | 97.47 | 93.53 | 60.29 | 63.93 | 92.01 | **0.98** | 98.52 |
| | Blend | 88.76 | 3.66 | 95.86 | 92.71 | 4.50 | 97.42 | 90.65 | **0.84** | 98.21 | 93.11 | 36.19 | 81.77 | 91.84 | 1.69 | **98.39** |
| | WaNet | 88.14 | 83.56 | 55.38 | 92.57 | 1.28 | 98.74 | 91.48 | 1.57 | 98.05 | 93.33 | 2.23 | 98.64 | 92.39 | **0.52** | 99.02 |
| | IAB | 90.21 | 8.37 | 93.54 | 93.01 | 1.01 | 98.62 | 90.23 | 0.58 | 97.45 | 93.24 | 0.98 | 98.75 | 92.37 | 0.38 | 98.62 |
| | Lira | 92.10 | 93.77 | 52.01 | 93.12 | 0.50 | 99.15 | 90.93 | 0.86 | 97.88 | **93.88** | 19.88 | 89.84 | 92.80 | **0.11** | 99.19 |
| | Average | 79.76 | 47.38 | 69.18 | 92.86 | 2.02 | **98.41** | 89.96 | **1.14** | 97.40 | 93.44 | 39.93 | 79.75 | 92.24 | 2.43 | 97.90 |

**GTSRB**

| | | No Defense | | | FP | | | CLP | | | ANP | | | RNP | | |
|---|---|---|---|---|---|---|---|---|---|---|---|---|---|---|---|---|
| | | ACC↑ | ASR↓ | DER↑ | ACC↑ | ASR↓ | DER↑ | ACC↑ | ASR↓ | DER↑ | ACC↑ | ASR↓ | DER↑ | ACC↑ | ASR↓ | DER↑ |
| | BadNets | 95.08 | 100.00 | – | **95.31** | 100.00 | 50.00 | 93.74 | 92.98 | 52.84 | 89.53 | 100.00 | 47.78 | 80.69 | 2.96 | 91.83 |
| | Trojan | 94.39 | 100.00 | – | **94.81** | 99.99 | 50.00 | 87.00 | 0.18 | 99.21 | 87.00 | 99.82 | 46.56 | 85.92 | 0.98 | 95.93 |
| | Blend | 93.85 | 99.50 | – | 94.68 | 97.84 | 50.83 | 92.26 | 99.41 | 49.25 | 82.53 | 96.87 | 45.80 | 93.45 | 83.75 | 57.82 |
| | WaNet | 93.99 | 97.07 | – | **95.76** | 10.79 | 93.14 | 20.59 | 100.00 | 13.30 | 84.39 | **0.00** | 95.16 | 87.75 | **0.00** | 96.84 |
| | IAB | 94.09 | 97.22 | – | 94.25 | 75.58 | 60.82 | 92.76 | 7.85 | 94.02 | 81.78 | 33.72 | 77.00 | 92.28 | **0.00** | **99.11** |
| | Lira | 93.97 | 99.91 | – | **94.18** | 7.99 | 95.96 | 92.16 | 11.32 | 93.39 | 79.94 | 25.95 | 79.96 | 85.52 | **0.00** | 95.73 |
| | Average | 94.23 | 98.95 | – | **94.83** | 65.36 | 66.79 | 80.75 | 51.96 | 67.00 | 84.20 | 59.39 | 65.38 | 87.60 | 14.45 | 89.54 |

| | | TSBD | | | FT-SAM | | | SAU | | | FST | | | Ours | | |
|---|---|---|---|---|---|---|---|---|---|---|---|---|---|---|---|---|
| | | ACC↑ | ASR↓ | DER↑ | ACC↑ | ASR↓ | DER↑ | ACC↑ | ASR↓ | DER↑ | ACC↑ | ASR↓ | DER↑ | ACC↑ | ASR↓ | DER↑ |
| GTSRB | BadNets | 94.81 | 100.00 | 49.86 | 95.15 | 100.00 | 50.00 | 94.37 | **0.00** | **99.64** | 89.49 | 100.00 | 47.20 | 94.27 | 6.78 | 96.20 |
| | Trojan | 93.44 | 99.99 | 49.53 | 94.25 | 43.53 | 78.17 | 92.26 | 0.09 | 98.89 | 90.04 | 95.40 | 50.13 | 93.40 | 0.49 | **99.27** |
| | Blend | 93.47 | 78.62 | 60.25 | 94.57 | 60.78 | 69.36 | 94.03 | **0.40** | **99.55** | 89.45 | 17.73 | 88.68 | 93.39 | 7.06 | 95.99 |
| | WaNet | 91.82 | 94.20 | 50.35 | 95.36 | 0.01 | 98.53 | 95.04 | 0.03 | 98.52 | 89.81 | 0.01 | 96.44 | 95.60 | **0.00** | 98.54 |
| | IAB | 94.24 | 6.96 | 95.13 | **94.48** | 0.23 | 98.50 | 94.30 | 0.02 | 98.60 | 90.47 | **0.00** | 96.80 | 94.21 | **0.00** | 98.61 |
| | Lira | 91.34 | 70.18 | 63.55 | 93.60 | **0.00** | **99.77** | 87.08 | 0.01 | 96.50 | 88.84 | 0.01 | 97.39 | 92.79 | **0.00** | 99.37 |
| | Average | 93.18 | 74.99 | 61.44 | 94.57 | 34.09 | 82.39 | 92.85 | **0.09** | **98.62** | 89.68 | 35.53 | 79.44 | 93.94 | 2.39 | 98.00 |

**TinyImageNet**

| | | No Defense | | | FP | | | CLP | | | ANP | | | RNP | | |
|---|---|---|---|---|---|---|---|---|---|---|---|---|---|---|---|---|
| | | ACC↑ | ASR↓ | DER↑ | ACC↑ | ASR↓ | DER↑ | ACC↑ | ASR↓ | DER↑ | ACC↑ | ASR↓ | DER↑ | ACC↑ | ASR↓ | DER↑ |
| | BadNets | 61.98 | 99.97 | – | 58.26 | 99.94 | 48.16 | 35.09 | 0.25 | 86.41 | 43.78 | 91.76 | 44.91 | 61.19 | 0.09 | **99.45** |
| | Trojan | 61.58 | 100.00 | – | 58.07 | 99.12 | 48.69 | 59.59 | 0.22 | 88.89 | 36.96 | 100.00 | 37.39 | 59.85 | **0.00** | 98.82 |
| | Blend | 62.28 | 99.97 | – | 57.68 | 0.43 | 97.47 | 54.29 | 9.52 | 91.23 | 28.03 | 93.91 | 35.96 | 60.91 | **0.01** | 99.35 |
| | WaNet | 62.37 | 99.58 | – | 58.80 | 0.17 | 97.92 | 48.17 | 0.63 | 92.37 | 36.74 | 99.05 | 37.75 | 36.42 | 51.66 | 61.28 |
| | IAB | 62.56 | 99.39 | – | 59.03 | 0.09 | 97.88 | 59.41 | 0.19 | 98.02 | 34.74 | 0.84 | 85.85 | 50.05 | 15.66 | 86.10 |
| | Lira | 62.19 | 99.99 | – | 58.87 | 0.32 | 98.17 | 58.79 | 0.24 | 98.17 | 41.04 | 99.99 | 39.42 | 54.19 | 0.00 | 95.99 |
| | Average | 62.16 | 99.82 | – | 58.45 | 33.35 | 81.38 | 52.56 | 1.84 | 94.18 | 36.88 | 80.92 | 46.88 | 53.77 | 11.24 | 90.17 |

| | | TSBD | | | FT-SAM | | | SAU | | | FST | | | Ours | | |
|---|---|---|---|---|---|---|---|---|---|---|---|---|---|---|---|---|
| | | ACC↑ | ASR↓ | DER↑ | ACC↑ | ASR↓ | DER↑ | ACC↑ | ASR↓ | DER↑ | ACC↑ | ASR↓ | DER↑ | ACC↑ | ASR↓ | DER↑ |
| TinyImageNet | BadNets | 50.58 | 29.31 | 79.63 | 52.96 | 0.23 | 95.36 | 52.95 | 0.54 | 95.20 | 53.58 | 30.51 | 80.53 | 56.33 | **0.00** | 97.16 |
| | Trojan | 50.58 | 0.22 | 94.39 | 52.29 | 0.26 | 95.22 | 52.07 | 0.35 | 95.07 | 51.37 | 0.14 | 94.82 | 56.71 | 0.01 | 97.36 |
| | Blend | 51.48 | 0.05 | 94.56 | 53.50 | 0.17 | 95.51 | 53.29 | 6.77 | 92.10 | 54.04 | 0.48 | 95.62 | 57.97 | 0.01 | 97.82 |
| | WaNet | 50.41 | 99.88 | 44.02 | 55.18 | 0.39 | 96.00 | 57.43 | 3.90 | 95.37 | 53.89 | 0.12 | 95.49 | 61.37 | 0.02 | **99.28** |
| | IAB | 51.69 | 83.85 | 52.33 | 55.91 | 0.36 | 96.19 | 55.36 | 1.73 | 95.23 | 54.57 | **0.04** | 95.68 | 61.12 | 2.39 | 99.78 |
| | Lira | 50.56 | 1.42 | 93.47 | 54.28 | 0.29 | 95.89 | 55.19 | 0.51 | 96.24 | 52.58 | 0.39 | 94.99 | **59.78** | 0.02 | **98.78** |
| | Average | 50.88 | 35.79 | 76.40 | 54.02 | **0.28** | 95.70 | 54.38 | 2.30 | 94.87 | 53.34 | 5.28 | 92.86 | **58.88** | 0.41 | **98.06** |

Table 3: Comparison of the backdoor removal results for CIFAR-10 on ResNet-50.

| | No Defense | | | FP | | | CLP | | | ANP | | | RNP | | |
|---|---|---|---|---|---|---|---|---|---|---|---|---|---|---|---|---|
| | ACC↑ | ASR↓ | DER↑ | ACC↑ | ASR↓ | DER↑ | ACC↑ | ASR↓ | DER↑ | ACC↑ | ASR↓ | DER↑ | ACC↑ | ASR↓ | DER↑ |
| BadNets | 91.48 | 100.00 | – | 91.02 | 58.88 | 70.33 | 63.83 | 9.33 | 81.51 | 15.44 | **4.40** | 59.82 | **91.10** | 100.00 | 49.85 |
| Trojan | 92.69 | 100.00 | – | 91.83 | 2.70 | **98.22** | 50.80 | 3.24 | 77.43 | 89.84 | 78.10 | 60.17 | 87.51 | 13.18 | 91.47 |
| Blend | 92.11 | 99.59 | – | 91.16 | 17.68 | 90.48 | 47.62 | 5.68 | 74.71 | 16.25 | 21.52 | 51.66 | 42.84 | 15.06 | 68.19 |
| WaNet | 92.83 | 98.92 | – | 91.99 | 0.90 | **98.59** | 59.54 | 0.14 | 82.74 | 12.36 | **0.00** | 60.48 | 68.84 | 90.16 | 43.64 |
| IAB | 92.68 | 98.71 | – | 91.93 | 1.28 | 98.34 | 56.43 | 20.67 | 70.90 | 15.08 | 16.37 | 53.66 | 18.18 | 26.02 | 50.38 |
| Lira | 91.40 | 100.00 | – | 90.42 | 0.33 | **99.34** | 24.27 | 19.53 | 56.67 | 53.88 | 29.34 | 66.57 | 37.11 | 1.76 | 71.98 |
| Average | 92.20 | 99.54 | – | 91.39 | 13.63 | 92.55 | 50.41 | 9.77 | 73.99 | 33.81 | 24.96 | 58.73 | 57.60 | 41.03 | 62.58 |

| | TSBD | | | FT-SAM | | | SAU | | | FST | | | Ours | | |
|---|---|---|---|---|---|---|---|---|---|---|---|---|---|---|---|---|
| | ACC↑ | ASR↓ | DER↑ | ACC↑ | ASR↓ | DER↑ | ACC↑ | ASR↓ | DER↑ | ACC↑ | ASR↓ | DER↑ | ACC↑ | ASR↓ | DER↑ |
| BadNets | 81.41 | 4.74 | 92.59 | 89.77 | 26.27 | 86.01 | 87.51 | 8.76 | 93.64 | 90.97 | 84.20 | 57.65 | 89.01 | 4.68 | **96.41** |
| Trojan | 87.27 | 2.24 | 96.17 | 91.10 | 3.21 | 97.60 | 88.22 | 1.16 | 97.19 | **92.07** | 15.66 | 91.86 | 90.01 | **1.10** | 98.11 |
| Blend | 83.42 | 5.07 | 92.92 | 90.00 | 6.08 | 95.70 | 88.60 | **1.36** | **97.36** | 91.18 | 15.83 | 91.41 | 89.19 | 3.10 | 96.78 |
| WaNet | 83.80 | 71.51 | 59.19 | 90.66 | 1.57 | 97.59 | 89.48 | 1.86 | 96.86 | **92.30** | 1.58 | 98.41 | 90.05 | 0.69 | 97.73 |
| IAB | 82.83 | 76.14 | 56.36 | 90.74 | 1.10 | 97.84 | 88.51 | 1.58 | 96.48 | **92.35** | 1.47 | **98.46** | 89.57 | **0.69** | 97.46 |
| Lira | 68.79 | 5.21 | 86.09 | 89.23 | 0.53 | 98.65 | 86.59 | 0.82 | 97.18 | **90.67** | 10.01 | 94.63 | 88.63 | **0.12** | 98.55 |
| Average | 81.25 | 27.49 | 80.55 | 90.25 | 6.46 | 95.56 | 88.15 | 2.59 | 96.45 | **91.59** | 21.46 | 88.74 | 89.40 | **1.73** | **97.51** |

$\max(0, \Delta\mathrm{ASR} + 1))/2$, which takes values in $[0, 1]$. A DER closer to 1 indicates that the backdoor is more effectively removed while preserving clean accuracy.

**Main Results.** Table 2 presents the backdoor removal results for various attack and defense methods on ResNet-18. Overall, our method achieves consistently superior DER across datasets. On Tiny-ImageNet, it attains the highest DER of 98.06%, with competitive ACC and ASR. On GTSRB, our approach yields a DER substantially higher than competing defenses, reflecting both strong attack suppression and accuracy preservation. On CIFAR-10, it achieves the best DER of 98.39% on Blend and remains competitive across other attacks. In addition, the results on CIFAR-10 with ResNet-50

are reported in Table 3. On ResNet-50, our method outperforms all defenses, including FT-SAM and SAU, which performed well on ResNet-18, in terms of both ACC and ASR. These results indicate that our approach suppresses backdoor success to near-zero while preserving high clean accuracy across diverse attack types, architectures, and datasets compared to the state-of-the-art existing defense methods.

## 6 CONCLUSION

In this work, we introduced a novel backdoor removal framework that reconstructs Trigger-Activated Changes (TAC) in the latent representation and leverages the reconstructed TAC for effective backdoor removal. Our method consists of two stages: recontructing TAC in the latent representation by computing minimal perturbations which misclassify any clean data into a target class for all classes and identifying the poisoned class via statistical test of extreme selection bias, and fine-tuning the model using the optimized perturbation of the poisoned class. Our experiments demonstrated that our method achieves superior backdoor suppression while maintaining high clean accuracy in any attack type, dataset, and architecture. As future work, we aim to extend our method to settings with multiple poisoned classes, since the current poisoned identification method assumes the perturbation norm of the single poisoned class is smaller than that of all other clean classes.

## ETHICS STATEMENT

Our work does not involve human participants, sensitive personal data, or experiments with potential risks to individuals or communities. We relied on publicly available datasets that are widely recognized in the research community, and we ensured ethical use of data by citing sources appropriately and complying with dataset licenses.

## REPRODUCIBILITY STATEMENT

The experimental configurations used for reproduction are described in Section 5.1 and Appendix C.

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

## APPENDIX

The Appendix provides additional technical details, extended discussions, and supplementary experimental results to support the main paper. Its structure is organized as follows:

- Appendix A presents a brief statement regarding the use of large language models during manuscript preparation.
- Appendix B provides additional related work on backdoor defenses, including fine-tuning–based approaches, training-stage defenses, and inference-stage detection methods.
- Appendix C summarizes implementation details for all backdoor attacks and defense baselines used in our experiments, expanding upon the configurations described in Section 5.

- Appendix D offers extended methodological details, including the complete algorithmic procedure, derivations of the dual optimization problem, feasibility analysis, and proofs of strong duality.
- Appendix E analyzes the computational complexity of our TAC reconstruction framework and presents empirical runtime measurements across datasets and model architectures.
- Appendix F contains additional experimental results, such as evaluations on large-scale datasets and models, further comparisons with prior neuron identification methods, ablation studies on poisoned class identification, pruning-based variants, sensitivity analyses for hyperparameters, experiments with varying reference dataset sizes, effectiveness under low poisoning rates, effectiveness against a defense-aware attack, and visualization of the reconstructed TAC.

## A   LLM Usage

While drafting this paper, we used a large language model (e.g., GPT-5) to assist with grammar correction, readability improvements, and literature searches. The scientific content, original ideas, and experimental findings are entirely the work of the authors.

## B   Additional Related Works for Backdoor Defenses

In Section 2.2, we discussed backdoor defenses that aim to remove backdoors by identifying backdoor neurons from compromised models, and here we introduce other defense strategies following the literature (Abbasi et al., 2025).

### B.1   Backdoor Removal without Backdoor Neuron Identification

Several recent defenses avoid explicitly identifying backdoor neurons and instead mitigate backdoors through fine-tuning and feature regularization. FT-SAM (Zhu et al., 2023) employs sharpness-aware minimization during fine-tuning to suppress backdoor-sensitive parameters. SAU (Wei et al., 2023) uses adversarial perturbations to unlearn shared backdoor features across classes, while FST (Min et al., 2023) adjusts feature distributions to shift poisoned representations away from decision boundaries. FIP (Karim et al., 2024) leverages Fisher information to purify representations and reduce the influence of backdoors.

### B.2   Training-Stage Defenses

Training-stage defenses aim to prevent the learning of backdoor correlations during model training by modifying optimization dynamics, restructuring the training process, or limiting the influence of poisoned data. A key insight is that poisoned data often behave differently from clean data in early training , e.g., faster loss reduction or more sensitive feature transformations, which can be exploited to detect and neutralize them.

Representative methods include Anti-Backdoor Learning (ABL) (Li et al., 2021), which isolates suspicious low-loss data in early epochs and later unlearns them to break trigger–label associations. Extensions refine this idea: Adaptively Splitting Dataset (ASD) (Gao et al., 2023) adaptively partitions data into clean and poisoned pools; Progressive Isolation (PIPD) (Chen et al., 2024) progressively reduces false positives in isolation; and Mind Control through Causal Inference (MCCI) (Hu et al., 2025) leverages causal modeling to disentangle triggers from true classes.

### B.3   Inference-Stage Defenses

Inference-stage defenses aim to identify or neutralize trigger-bearing inputs during inference, making them especially useful when retraining or model inspection is impractical. A representative approach is perturbation-based detection, where methods such as STRIP (Gao et al., 2019) perturb incoming inputs and measure the entropy of predictions; consistently low entropy often indicates the presence of a trigger. Another line focuses on input purification, with Februus (Doan et al., 2020)

---

**Algorithm 1** Backdoor Removal via Reconstructing TAC in the Latent Representation

---

**Require:** Compromised model parameter $\boldsymbol{\theta}_{\mathrm{bd}}$, a reference dataset $D_{\mathrm{ref}}$, number of trials $T_{\mathrm{pci}}$, tolerance $\eta$, significance level $\alpha$, hyperparameter $\beta$.

**Ensure:** Fine-tuned model $\boldsymbol{\theta}_{\mathrm{ft}}^*$

1: **Phase 1: Reconstructing TAC in the Latent Representation**
2:   $B_{\mathrm{LR}} = \{\phi_{\boldsymbol{\theta}_{\mathrm{bd}}}(\boldsymbol{x}_1), \phi_{\boldsymbol{\theta}_{\mathrm{bd}}}(\boldsymbol{x}_2), \cdots, \phi_{\boldsymbol{\theta}_{\mathrm{bd}}}(\boldsymbol{x}_{|D_{\mathrm{ref}}|})\}$
3:   **Initialization:** Set $N_k \leftarrow 0$ for all classes $k \in [C]$.
4:   **for** $i = 1, \ldots, T_{\mathrm{pci}}$ **do**
5:     Sample a subset $B^{(i)} \subset B_{\mathrm{LR}}$.
6:     Compute perturbations $\{\boldsymbol{s}_k^{*(i)}\}_{k=1}^C$ by solving the optimization in Section 4.1.
7:     Let $c^{(i)} = \arg\min_k \|\boldsymbol{s}_k^{*(i)}\|_2$.
8:     Update $N_{c^{(i)}} \leftarrow N_{c^{(i)}} + 1$.
9:   **end for**
10:   Compute $N^* = \max_k N_k$ and $p_{\mathrm{max}} = \frac{N^*}{T_{\mathrm{pci}}}$.

11:   **Hypothesis Test:**
12:   Consider the one-sided test

$$H_0 : p_{\mathrm{max}} \leq \eta \quad \text{vs.} \quad H_1 : p_{\mathrm{max}} > \eta.$$

13:   Under the least favorable clean case, evaluate

$$p_{\mathrm{val}} = \Pr[X \geq N^*], \quad X \sim \mathrm{Binomial}(T_{\mathrm{pci}}, \eta).$$

14:   **if** $p_{\mathrm{val}} < \alpha$ **then**
15:     Declare the model compromised.
16:     Identify the poisoned class as $t = \arg\max_k N_k$.
17:   **else**
18:     Conclude the model is consistent with being clean and return None.
19:   **end if**
20:   After detection, compute $\boldsymbol{s}_t^*$ using the full reference dataset.
21: **Phase 2: Backdoor Removal with the Perturbation in the Poisoned Class**
22:   Fine-tune the compromised model by solving
23:

$$\boldsymbol{\theta}_{\mathrm{ft}}^* = \arg\min_{\boldsymbol{\theta}_{\mathrm{bd}}} \frac{1}{|D_{\mathrm{ref}}|} \sum_{i=1}^{|D_{\mathrm{ref}}|} \left[ \ell(f(\boldsymbol{x}_i; \boldsymbol{\theta}_{\mathrm{bd}}), \boldsymbol{y}_i) + \beta\, \ell(\mathrm{Softmax}(\boldsymbol{W}_L^\top(\hat{\boldsymbol{x}}_i + \boldsymbol{s}_t^*) + \boldsymbol{b}), \boldsymbol{y}_i) \right].$$

24: **Return** $\boldsymbol{\theta}_{\mathrm{ft}}^*$

---

removing suspicious regions through inpainting to recover benign content and mitigate patch-style trojans.

Beyond perturbation and purification, interpretability-based defenses such as SentiNet (Chou et al., 2020) leverage saliency maps to localize highly influential regions and assess their generalization across inputs, enabling detection of physical-world triggers. Similarly, TeCo (Liu et al., 2023) exploits robustness discrepancies under common image corruptions, showing that poisoned inputs behave inconsistently compared to clean ones, thus allowing detection without soft classes or auxiliary clean datasets. More recent studies, including CBD (Xiang et al., 2023), TED (Mo et al., 2023), and BaDExpert (Xie et al., 2023), further enhance detection reliability by leveraging statistical probability bounds, topological dynamics, or explicit extraction of backdoor functionality. As another line of work, REFINE (Chen et al., 2025) introduces a model reprogramming strategy that jointly employs an input transformation module and an output remapping module. By aggressively transforming inputs while simultaneously remapping output classes, REFINE reduces the effectiveness of triggers without severely degrading clean accuracy.

## C  IMPLEMENTATION DETAILS

We conducted experiments based on the implementation in the `OrthogLinearBackdoor` (Zhang et al., 2024) repository [1].

### C.1  BACKDOOR ATTACKS

We implemented six representative backdoor attack methods. Default configurations of all attacks follow the `OrthogLinearBackdoor`. As described in Section 5.1, all attacks were trained for 100 epochs using stochastic gradient descent (SGD) with a learning rate of 0.1 and cosine annealing as the learning rate scheduler.

- **BadNets** (Gu et al., 2019). A patch-based backdoor that stamps a fixed visible pattern onto inputs to induce a target class; in our experiments we follow existing work (Zhang et al., 2024) and use the sunflower image as the trigger.
- **Trojan** (Liu et al., 2018b). A trigger-stamping attack which plants a small image-based trigger; here the trigger is a small sunflower image with a transparent background.
- **Blend** (Chen et al., 2017). A blending-style attack that mixes a trigger image into the entire input with a given transparency; we use a Hello-Kitty image blended at an alpha of 0.2.
- **WaNet** (Nguyen & Tran, 2021). A warping-based backdoor that applies imperceptible geometric distortions (image warps) as the trigger, producing stealthy, input-agnostic perturbations.
- **IAB** (Nguyen & Tran, 2020). An input-dependent attack that generates a dynamic trigger conditioned on each input, making detection and removal more challenging.
- **Lira** (Doan et al., 2021b). A backdoor attack generating learnable, imperceptible, and robust triggers, making them hard to detect and defend.

### C.2  BACKDOOR DEFENSES

We implemented eight backdoor removal methods. Unless otherwise specified, implementations are based on the `OrthogLinearBackdoor` repository, while methods without public implementations were re-implemented following the authors' original repositories or BackdoorBench (Wu et al., 2022) which is another benchmark framework that provides unified implementations of representative backdoor attacks and defenses for fair and reproducible evaluation.

- **Fine-Pruning (FP)** (Liu et al., 2018a). This method prunes neurons that are inactive on clean data, assuming such neurons are likely backdoor-related. We set the pruning ratio as 0.2, fine-tuning epochs as 50, the optimizer as SGD, learning rate as 0.01 and learning rate scheduler as cosine annealing.
- **Channel Lipschitzness Pruning (CLP)** (Zheng et al., 2022). CLP removes channels with abnormally large Lipschitz constants, aiming to suppress backdoor activations. The implementation is not included in the `OrthogLinearBackdoor` repository, we refer the implementation in BackdoorBench (Wu et al., 2022). We also set the threshold parameter as 3.0 following the original paper (Zheng et al., 2022).
- **Adversarial Neuron Pruning (ANP)** (Wu & Wang, 2021). ANP identifies and prunes neurons that are highly sensitive to adversarial perturbations. In our experiments, for CIFAR-10 we set $\epsilon = 0.3, \alpha = 0.2$ and the pruning threshold as 0.2; for GTSRB $\epsilon = 0.4, \alpha = 0.2$ and the pruning threshold as 0.4; and for TinyImageNet $\epsilon = 0.2, \alpha = 0.3$ and the pruning threshold 0.001 where $\epsilon$ and $\alpha$ are the hyperparameters introduced in the original paper.
- **Reconstructive Neuron Pruning (RNP)** (Li et al., 2023). RNP prunes neurons whose removal minimally affects the reconstruction of clean representations from the unlearned model. The implementation is not included in the `OrthogLinearBackdoor` repository, we refer the implementation in BackdoorBench. We set the pruning threshold as 0.7 for CIFAR-10, the pruning threshold as 0.95 for GTSRB, and the pruning threshold as 0.1 for TinyImageNet.

---

[1] `https://github.com/KaiyuanZh/OrthogLinearBackdoor/blob/main/`

- **Two-Stage Backdoor Defense (TSBD)** (Lin et al., 2024). TSBD identifies backdoor neuron based on Neuron Weight Change (NWC) which is the difference between the compromised model's weights and the unlearned model's weights, and conducts activeness-aware fine-tuning to mitigate backdoors. The implementation is not included in the `OrthogLinearBackdoor` repository, we refer the implementation in the original paper (Lin et al., 2024). Following the original paper, after calculating NWC, we selected 15% of the top-neurons and pruned 70% of the top-subweights within them. In our experiments, we attempted to use activeness-aware fine-tuning following the original paper, but since the accuracy dropped significantly after fine-tuning, we instead adopted standard fine-tuning. Fine-tuning configuration in TSBD is the same as that of FP.

- **FT-SAM** (Zhu et al., 2023). This method leverages sharpness-aware minimization (SAM) during fine-tuning to suppress backdoor behaviors. The implementation is not included in the `OrthogLinearBackdoor` repository, we refer the implementation in Backdoor-Bench. The training configuration and hyperparameters are followed as BackdoorBench.

- **Shared Adversarial Unlearning (SAU)** (Wei et al., 2023). SAU uses adversarial perturbations to unlearn shared backdoor features across classes. The implementation is not included in the `OrthogLinearBackdoor` repository, we refer the implementation in BackdoorBench. The training configuration and hyperparamters are followed as BackdoorBench.

- **Feature Shift Tuning (FST)** (Min et al., 2023). FST fine-tunes models by aligning feature distributions to shift away backdoor-related representations. The implementation is not included in the `OrthogLinearBackdoor` repository, we refer the implementation in the original paper (Min et al., 2023). The hyperparameter that balances the loss terms (denoted as $\alpha$ in the original paper) is set to 0.2 for CIFAR-10, 0.1 for GTSRB, and 0.001 for TinyImageNet, following the original paper.

## D DETAILS OF OUR PROPOSED METHOD

### D.1 ALGORITHMS

To clarify our proposed method as described in Section 4, we present the detailed procedure in Algorithm 1.

### D.2 DERIVATION PROCESS FOR DUAL PROBLEM

We describe the derivation process from equation 4 to equation 5.

**Lagrangian and dual function.** Introduce the dual variable $\boldsymbol{\lambda} \in \mathbb{R}^{C-1}$ with $\boldsymbol{\lambda} \geq \mathbf{0}$ for the inequality constraints from equation 4. The Lagrangian is

$$\mathcal{L}(\boldsymbol{s}_k, \boldsymbol{\lambda}) = \frac{1}{2} \|\boldsymbol{s}_k\|_2^2 - \boldsymbol{\lambda}^\top (\boldsymbol{V}_k \boldsymbol{s}_k - \boldsymbol{m}) \quad \text{s.t.} \quad \boldsymbol{\lambda} \geq \mathbf{0}.$$

The dual function is obtained by minimizing the Lagrangian over the primal variable:

$$g(\boldsymbol{\lambda}) = \inf_{\boldsymbol{s}_k} \mathcal{L}(\boldsymbol{s}_k, \boldsymbol{\lambda}).$$

Stationarity (optimality in $\boldsymbol{s}_k$) gives

$$\nabla_{\boldsymbol{s}_k} \mathcal{L}(\boldsymbol{s}_k, \boldsymbol{\lambda}) = \boldsymbol{s}_k - \boldsymbol{V}_k^\top \boldsymbol{\lambda} = \mathbf{0} \implies \boldsymbol{s}_k = \boldsymbol{V}_k^\top \boldsymbol{\lambda}.$$

Plugging this into $\mathcal{L}$ yields

$$g(\boldsymbol{\lambda}) = \boldsymbol{\lambda}^\top \boldsymbol{m} - \frac{1}{2} \|\boldsymbol{V}_k^\top \boldsymbol{\lambda}\|_2^2.$$

Therefore, the dual problem is the concave maximization

$$\boldsymbol{\lambda}^* = \underset{\boldsymbol{\lambda}}{\operatorname{argmax}} \ \boldsymbol{\lambda}^\top \boldsymbol{m} - \frac{1}{2} \|\boldsymbol{V}_k^\top \boldsymbol{\lambda}\|_2^2 \quad \text{s.t.} \quad \boldsymbol{\lambda} \geq \mathbf{0}.$$

### D.3 FEASIBLE SOLUTION

**Theorem 1.** *If $C - 1 < d_{\mathrm{emb}}$ and $\boldsymbol{V}_k$ has full row rank, i.e. $\mathrm{rank}(\boldsymbol{V}_k) = C - 1$, then the primal problem equation 4 has a feasible solution.*

*Proof.* By Farkas' lemma (Boyd & Vandenberghe, 2004), exactly one of the following two statements holds:

    1. There exists $\boldsymbol{s}_k \in \mathbb{R}^{d_{\mathrm{emb}}}$ such that $\boldsymbol{V}_k \boldsymbol{s}_k \geq \boldsymbol{m}$.

    2. There exists $\boldsymbol{\lambda} \in \mathbb{R}^{C-1}$ such that $\boldsymbol{V}^\top \boldsymbol{\lambda} = \boldsymbol{0}, \quad \boldsymbol{\lambda} \geq \boldsymbol{0}, \quad \boldsymbol{m}^\top \boldsymbol{\lambda} < 0$.

If $\mathrm{rank}(\boldsymbol{V}_k) = C - 1$ with $C - 1 < d_{\mathrm{emb}}$, then $\ker(\boldsymbol{V}_k^\top) = \{\boldsymbol{0}\}$. Hence the only $\boldsymbol{\lambda}$ satisfying $\boldsymbol{V}_k^\top \boldsymbol{\lambda} = \boldsymbol{0}$ is $\boldsymbol{\lambda} = \boldsymbol{0}$, which cannot yield $\boldsymbol{m}^\top \boldsymbol{\lambda} < 0$. Thus (2) is impossible, and therefore (1) must hold. Hence, there exists $\boldsymbol{s}_k$ with $\boldsymbol{V}_k^\top \boldsymbol{s}_k \geq \boldsymbol{m}$, and the primal problem is feasible. $\square$

### D.4 STRONG DUALITY

**Theorem 2.** *If $C - 1 < d_{\mathrm{emb}}$ and $\mathrm{rank}(\boldsymbol{V}_k) = C - 1$, then the primal problem equation 4 and equation 5 satisfy strong duality.*

*Proof.* To establish this result, we show that the primal problem is a convex optimization problem and that it satisfies Slater's condition.

**1. Convexity.** The objective $\frac{1}{2}\|\boldsymbol{s}_k\|_2^2$ is strongly convex. The feasible region is given by

$$Z := \{\boldsymbol{s}_k \in \mathbb{R}^{d_{\mathrm{emb}}} : \boldsymbol{V}_k^\top \boldsymbol{s}_k \geq \boldsymbol{m}\} = \bigcap_{i=1}^{C-1} \{\boldsymbol{s}_k : \boldsymbol{v}_i^\top \boldsymbol{s}_k \geq m_i\},$$

where each set $\{\boldsymbol{s}_k : \boldsymbol{v}_i^\top \boldsymbol{s}_k \geq m_i\}$ is a half-space and therefore convex. Since the feasible set $Z$ is the intersection of convex sets, it is also convex. Thus, the problem equation 4 is a convex optimization problem.

**2. Slater's condition.** Since $\boldsymbol{V}_k$ has full row rank, we have $\mathrm{rank}(\boldsymbol{V}_k) = C - 1$. This implies that the linear map

$$J : \mathbb{R}^{d_{\mathrm{emb}}} \to \mathbb{R}^{C-1}, \qquad J(\boldsymbol{s}_k) = \boldsymbol{V}_k^\top \boldsymbol{s}_k,$$

is surjective. Hence, for any $\epsilon > 0$, there exists $\bar{\boldsymbol{s}}_k \in \mathbb{R}^{d_{\mathrm{emb}}}$ such that

$$\boldsymbol{V}_k^\top \bar{\boldsymbol{s}}_k = \boldsymbol{m} + \epsilon \boldsymbol{1}_{C-1}.$$

Since $\epsilon > 0$, it follows that

$$\boldsymbol{V}_k^\top \bar{\boldsymbol{s}}_k = \boldsymbol{m} + \epsilon \boldsymbol{1}_{C-1} > \boldsymbol{m},$$

which means that $\bar{\boldsymbol{s}}_k$ strictly satisfies all inequality constraints. In other words, $\bar{\boldsymbol{s}}_k \in \mathrm{relint}(Z)$, where $Z = \{\boldsymbol{s}_k \in \mathbb{R}^{d_{\mathrm{emb}}} : \boldsymbol{V}_k^\top \boldsymbol{s}_k \geq \boldsymbol{m}\}$ and $\mathrm{relint}(Z)$ means the relative interior of the set $Z$. Therefore, Slater's condition holds for problem equation 4.

$\square$

## E COMPUTATIONAL COMPLEXITY AND RUNTIME ANALYSIS OF TAC RECONSTRUCTION

This section provides a detailed analysis of the computational efficiency of the TAC reconstruction procedure introduced in Section 4.1, together with empirical runtime measurements across datasets and architectures. The goal is to clarify that the proposed optimization is computationally lightweight and practical even for large-scale classification settings.

Table 4: Mean and standard deviation of QP solving time (sec) for one class.

|  | ResNet-18 ($d_{\text{emb}} = 512$) | ResNet-50 ($d_{\text{emb}} = 2048$) | ViT-B/32 ($d_{\text{emb}} = 512$) |
|---|---|---|---|
| CIFAR-10 | $0.015388 \pm 0.000639$ | $0.032364 \pm 0.003099$ | $0.015485 \pm 0.000380$ |
| GTSRB | $0.032178 \pm 0.004127$ | $0.104500 \pm 0.002913$ | $0.033923 \pm 0.003504$ |
| TinyImageNet | $0.141203 \pm 0.002386$ | $0.474484 \pm 0.021590$ | $0.142614 \pm 0.004958$ |
| ImageNet-1K | - | - | $1.170394 \pm 0.018746$ |

Table 5: Backdoor removal results for TinyImageNet and ImageNet-1K.

|  |  | No Defense | | | FT-SAM | | | SAU | | | FST | | | Ours | | |
|---|---|---|---|---|---|---|---|---|---|---|---|---|---|---|---|---|
|  |  | ACC↑ | ASR↓ | DER↑ | ACC↑ | ASR↓ | DER↑ | ACC↑ | ASR↓ | DER↑ | ACC↑ | ASR↓ | DER↑ | ACC↑ | ASR↓ | DER↑ |
| TinyImageNet | BadNets | 62.76 | 100.00 | – | 60.70 | 100.00 | 48.97 | 53.35 | 6.16 | 92.21 | 39.13 | 2.16 | 87.10 | 59.99 | 7.32 | 94.96 |
|  | Trojan | 61.87 | 100.00 | – | 58.94 | 100.00 | 48.54 | 52.72 | 0.01 | 95.42 | 35.11 | 38.74 | 67.25 | 59.33 | 0.48 | 98.49 |
|  | Blend | 62.37 | 99.99 | – | 60.04 | 99.91 | 48.88 | 53.92 | 0.03 | 95.75 | 35.54 | 59.87 | 56.65 | 58.22 | 0.00 | 97.92 |
|  | Average | 62.33 | 99.99 | – | 59.89 | 99.97 | 48.79 | 53.33 | 2.07 | 94.46 | 36.59 | 33.59 | 70.33 | 59.18 | 2.60 | 97.12 |
| ImageNet-1K | BadNets | 70.52 | 100.00 | – | 69.01 | 99.996 | 49.25 | 51.16 | 0.28 | 90.18 | 58.54 | 0.01 | 94.00 | 59.54 | 0.00 | 94.51 |
|  | Trojan | 71.33 | 100.00 | – | 70.15 | 99.95 | 49.43 | 51.40 | 0.01 | 90.03 | 59.83 | 0.00 | 94.25 | 60.52 | 0.00 | 94.59 |
|  | Blend | 71.48 | 99.98 | – | 70.11 | 99.57 | 49.53 | 52.36 | 0.01 | 90.43 | 59.38 | 0.00 | 93.94 | 60.42 | 0.01 | 94.46 |
|  | Average | 71.11 | 99.99 | – | 69.76 | 99.84 | 49.40 | 51.64 | 0.10 | 90.21 | 59.25 | 0.00 | 94.06 | 60.16 | 0.00 | 94.52 |

## E.1 COMPUTATIONAL COMPLEXITY

We first describe the computational complexity of the quadratic problem (QP) we solve for per class, as defined in Equation (4):

$$\boldsymbol{\lambda}^* = \underset{\boldsymbol{\lambda}}{\arg\max} \quad \boldsymbol{\lambda}^\top \boldsymbol{m} - \frac{1}{2}\|\boldsymbol{V}_k^\top \boldsymbol{\lambda}\|_2^2 \quad \text{s.t.} \quad \boldsymbol{\lambda} \geq \boldsymbol{0}.$$

The dominant term in evaluating the dual objective is the computation of $\|\boldsymbol{V}^\top \lambda\|_2^2 = \boldsymbol{\lambda}^\top(\boldsymbol{V}\boldsymbol{V}^\top)\boldsymbol{\lambda} = (\boldsymbol{V}^\top\lambda)^\top(\boldsymbol{V}^\top\lambda)$, where $\boldsymbol{V} \in \mathbb{R}^{(C-1)\times d*\text{emb}}$. The matrix vector product $\boldsymbol{V}^\top\boldsymbol{\lambda}$ requires $\mathcal{O}(d_{\text{emb}}C)$ operations, and the linear term $\boldsymbol{\lambda}^\top\boldsymbol{m}$ adds $\mathcal{O}(C)$. Thus, each iteration of the convex solver costs: $\mathcal{O}(d_{\text{emb}}C)$. If the solver performs $T_{\text{solver}}$ iterations, the per-class cost becomes: $\mathcal{O}(T_{\text{solver}}d_{\text{emb}}C)$.

Importantly, this complexity depends only on the dimension $d_{\text{emb}}$ and the number of classes $C$; it does not depend on the size of the dataset or network depth. Moreover, the dimension is typically modest (e.g., 512 or 2048 for ResNet-based models), which keeps the optimization efficient even for large-scale classification problems.

## E.2 EMPIRICAL RUNTIME MEASUREMENTS

To supplement the complexity analysis, we also report actual time measurements for computing a minimal perturbation for per class in the following Table 4. Note that only the CPU, not the GPU, was used when computing the QP.

As a result, we confirmed that the solving time for a single class is very short and practical. Even for ImageNet-1K, which contains 1,000 classes, the computation runs fast—approximately 1.1 seconds for $d_{\text{emb}} = 512$.

Our poisoned class identification method requires additional computations: specifically, $C$ runs for the number of classes and $T_{\text{pci}}$ runs for the statistically necessary number of trials. A straightforward implementation would require $CT_{\text{pci}}$ additional computations, but since each computation is completely independent, they can be executed in parallel.

As a result, both the complexity analysis and the runtime measurements show that the proposed TAC reconstruction method is computationally lightweight and practical, even when scaling to larger numbers of classes or higher dimension of the latent representation.

# F    ADDITIONAL EXPERIMENTS

## F.1    RESULTS FOR LARGE-SCALE MODEL AND DATASET

To confirm the effectiveness of our proposed method on large-scale models datasets, we additionally conducted experiments on ViT-B/32 using TinyImageNet and ImageNet-1K as shown in Table 5. For the attack setups, we fine-tune the pre-trained models starting from publicly available pretrained ViT-B/32 checkpoints for 5 epochs with a learning rate of 0.0001. For our defense setup about TinyImageNet and ImageNet-1k, we fine-tune the compromised model for 25 epochs with a learning rate of 0.0005 and $\beta = 0.5$. We also compared our backdoor removal results with FT-SAM (Zhu et al., 2023), SAU (Wei et al., 2023), and FST (Min et al., 2023).

For TinyImageNet, our method consistently achieves a favorable balance between ACC and ASR. For instance, while SAU and FST achieve low ASR for certain attacks, their ACCs degrade substantially, dropping to 53% and 36% on average, respectively. In contrast, our method maintains a high ACC of 59.18% comparable to FT-SAM, while achieving a much lower ASR of 2.60%, far outperforming FT-SAM whose ASR remains above 99%. These findings indicate that, even in a large-scale model setting, the proposed method retains its effectiveness.

We further extended our evaluation to ImageNet-1K, which is an order of magnitude more challenging, involving 1000 classes. Remarkably, our method continues to perform strongly. Across BadNets, Trojan, and Blend attacks, our method reduces ASR to an average of 0.00%, matching or improving upon SAU and FST, both of which also achieve low ASR in this setting. Crucially, however, our method maintains ACC of 60.16%, which is dramatically higher than SAU (51.64%) and substantially higher than FST (59.25%). Meanwhile, FT-SAM remains ineffective on ViT-B/32, with ASR exceeding 99% across all three attacks. These ImageNet-1K results demonstrate that our method scales gracefully even in extremely large-scale dataset settings and remains robust.

Finally, these experiments provide strong evidence that the proposed defense generalizes beyond ResNet-based models and small datasets. Its effectiveness on ViT-B/32, combined with stable performance on both TinyImageNet and ImageNet-1K, confirms that the TAC reconstruction and fine-tuning mechanism does not rely on CNN-specific structures and remains robust even when applied to high-capacity architectures and large, complex datasets. This further reinforces the practicality and universality of our defense in real-world scenarios where large models and large-scale datasets are standard.

## F.2    COMPARISON WITH PREVIOUS METHODS FOR IDENTIFYING BACKDOOR NEURONS

We compare how accurately the perturbations in the latent representation obtained by our method can identify TAC-based backdoor neurons relative to existing approaches. Figure 4, Figure 5 and Figure 6 show the overlap rate with TAC-based backdoor neurons in the latent representation at the Top-$K$% for each dataset. These results show that among existing methods, RNP exhibits relatively stable performance, achieving high TAC coverage at small $K$ on CIFAR-10 and GTSRB, whereas TAC coverage at small $K$ on TinyImageNet shows low. In contrast, our proposed method consistently attains high TAC coverage at small $K$ across all datasets, demonstrating its stability and effectiveness in reconstructing TAC in the latent representation.

## F.3    EFFECTIVENESS OF POISONED CLASS IDENTIFICATION METHOD ON FINE-TUNING RESULTS

To verify the effectiveness of the poisoned class identification method, we conduct an ablation study in which fine-tuning is performed without identifying the poisoned class. Namely, we fine-tune a compromised model using the perturbations of all classes. Specifically, instead of applying $s_p^*$ in Equation (7) for our method, we randomly select $s^*$ from the set of perturbations at each training iteration for fine-tuning. The training configuration is the same as that of our method.

As shown in Table 6, even without poisoned class identification, ASR generally decreases to a level comparable to our proposed method although ASR of 17.26% remains for IAB on TinyImageNet and ASR of 20.23% for Blend on GTSRB. This is likely because, during training, the randomly selected $s^*$ occasionally corresponds to $s_p^*$. On the other hand, in terms of ACC, our method achieves higher

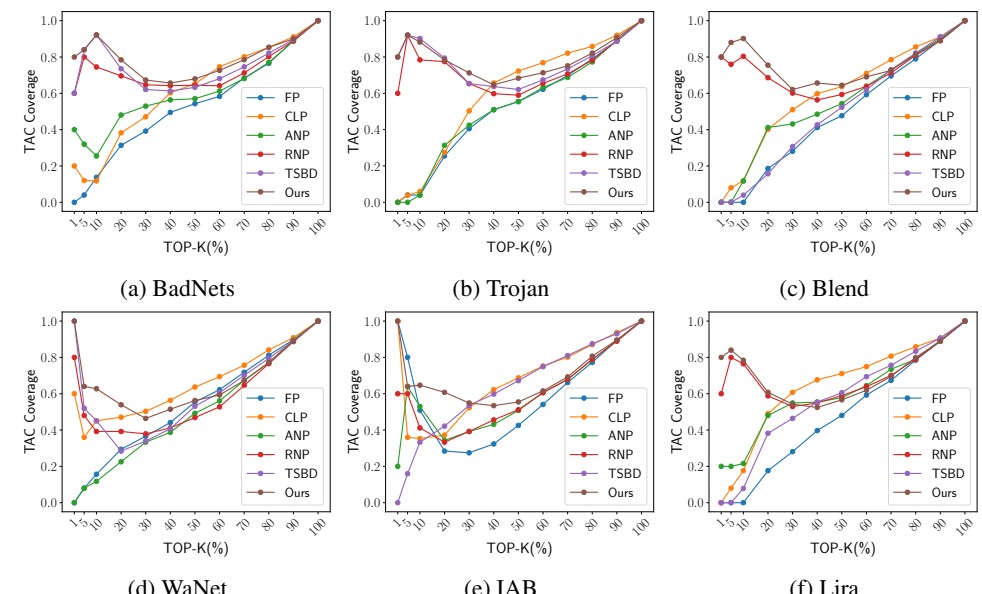

Figure 4: TAC coverage, defined as the overlap ratio between TAC-based backdoor neurons and those identified by each defense method on CIFAR-10.

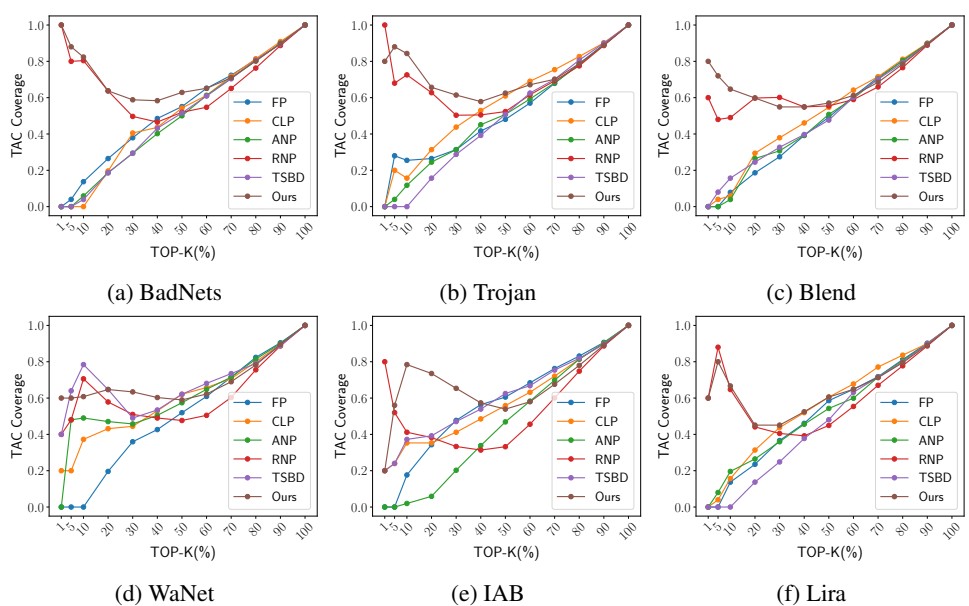

Figure 5: TAC coverage on GTSRB.

performance on CIFAR-10 and TinyImageNet. These results indicate that by leveraging only the perturbation of the poisoned class through the poisoned class identification method, our method is able to maintain higher accuracy while effectively removing backdoors.

### F.4 COMPARISON WITH PRUNING-BASED METHODS VIA RECONSTRUCTING TAC IN THE LATENT REPRESENTATION

As described in Section 4.3, we removed backdoors by reconstructing TAC with fine-tuning. Alternatively, pruning-based methods can provide another approach that leverages the reconstructed TAC for backdoor removal. Therefore, we further compare our method with pruning-based ap-

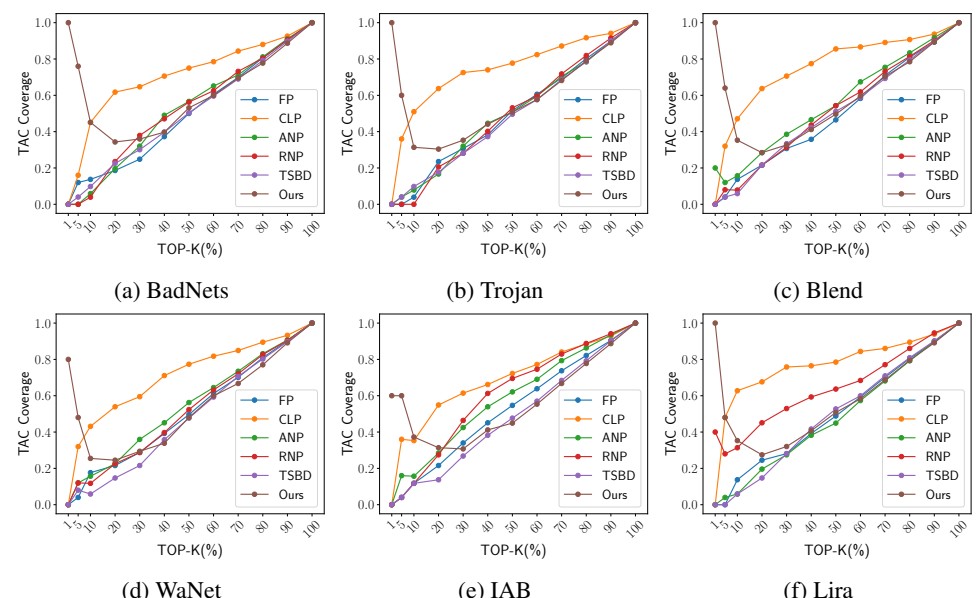

Figure 6: TAC coverage on TinyImageNet.

Table 6: Comparison of backdoor removal results between fine-tuning with the perturbations for all classes and fine-tuning with the perturbation of the poisoned class (Ours). "No PCI" means fine-tuning without the poisoned class identification (PCI) method.

| | | No Defense | | | No PCI | | | Ours | | |
|---|---|---|---|---|---|---|---|---|---|---|
| | | ACC↑ | ASR↓ | DER↑ | ACC↑ | ASR↓ | DER↑ | ACC↑ | ASR↓ | DER↑ |
| CIFAR-10 | BadNets | 93.81 | 100.00 | - | 90.91 | 16.13 | 90.48 | **92.03** | **10.88** | **93.67** |
| | Trojan | 94.00 | 100.00 | - | 90.36 | 2.67 | 96.85 | **92.01** | **0.98** | **98.52** |
| | Blend | 93.29 | 99.91 | - | 90.11 | 4.81 | 95.96 | **91.84** | **1.69** | **98.39** |
| | WaNet | 93.41 | 99.59 | - | 91.23 | 2.18 | 97.62 | **92.39** | **0.52** | **99.02** |
| | IAB | 93.57 | 98.81 | - | 90.38 | 1.89 | 96.87 | **92.37** | **0.38** | **98.62** |
| | Lira | 94.29 | 99.98 | - | 90.72 | 0.68 | 97.86 | **92.80** | **0.11** | **99.19** |
| | Average | 93.73 | 99.71 | - | 90.62 | 4.73 | 95.94 | **92.24** | **2.43** | **97.90** |
| GTSRB | BadNets | 95.08 | 100.00 | - | **94.89** | 7.68 | 96.07 | 94.27 | **6.78** | **96.20** |
| | Trojan | 94.39 | 100.00 | - | 93.08 | **0.41** | 99.14 | **93.40** | 0.49 | **99.27** |
| | Blend | 93.85 | 99.50 | - | 92.99 | 17.26 | 90.69 | **93.39** | **7.06** | **95.99** |
| | WaNet | 93.99 | 97.07 | - | **95.65** | 0.03 | 98.52 | 95.60 | **0.00** | **98.54** |
| | IAB | 94.09 | 97.22 | - | **94.36** | 0.02 | 98.60 | 94.21 | **0.00** | **98.61** |
| | Lira | 93.97 | 99.91 | - | 92.44 | 0.06 | 99.16 | **92.79** | **0.00** | **99.37** |
| | Average | 94.23 | 98.95 | - | 93.90 | 4.24 | 97.03 | **93.94** | **2.39** | **98.00** |
| TinyImageNet | BadNets | 61.98 | 99.97 | - | 53.17 | 0.25 | 95.45 | **56.33** | **0.00** | **97.16** |
| | Trojan | 61.58 | 100.00 | - | 52.09 | 0.28 | 95.11 | **56.71** | **0.01** | **97.56** |
| | Blend | 62.28 | 99.97 | - | 53.19 | 0.16 | 95.36 | **57.97** | **0.01** | **97.82** |
| | WaNet | 62.37 | 99.58 | - | 54.82 | 2.43 | 94.80 | **61.37** | **0.02** | **99.28** |
| | IAB | 62.56 | 99.39 | - | 55.57 | 20.23 | 86.08 | **61.12** | **2.39** | **97.78** |
| | Lira | 62.19 | 99.99 | - | 54.00 | 0.26 | 95.77 | **59.78** | **0.02** | **98.78** |
| | Average | 62.16 | 99.82 | - | 53.81 | 3.94 | 93.76 | **58.88** | **0.41** | **98.06** |

proaches by reconstructing TAC in the latent representation. As shown in Table 7, pruning alone can partially reduce ASR, but a considerable portion of backdoors remains (e.g., ASR of 69.89% for Trojan on CIFAR-10 and 56.08% for Blend on GTSRB), indicating that pruning itself is insufficient to completely eliminate the attacks. When combined with fine-tuning (Pruning+FT), the accuracy can be preserved, but the fine-tuning process often revives backdoors, leading to higher ASR in several cases (e.g., BadNets on CIFAR-10 where ASR returns to 100%). In contrast, our method consistently decreases ASR across all attack settings while preserving high accuracy. These results highlight that our approach overcomes the limitations of pruning-based methods and provides a more reliable defense against backdoor attacks.

Table 7: Comparison of backdoor removal results between pruning-based methods and fine-tuning-based method (Ours). Pruning ratio and fine-tuning configuration are set to be the same as those of FP.

| | | No Defense | | | Pruning | | | Pruning+FT | | | Ours | | |
|---|---|---|---|---|---|---|---|---|---|---|---|---|---|
| | | ACC↑ | ASR↓ | DER↑ | ACC↑ | ASR↓ | DER↑ | ACC↑ | ASR↓ | DER↑ | ACC↑ | ASR↓ | DER↑ |
| CIFAR-10 | BadNets | 93.81 | 100.00 | - | 88.87 | 37.74 | 78.66 | **93.66** | 100.00 | 49.92 | 92.03 | **10.88** | **93.67** |
| | Trojan | 94.00 | 100.00 | - | 90.26 | 69.89 | 63.19 | **93.55** | 97.86 | 50.85 | 92.01 | **0.98** | **98.52** |
| | Blend | 93.29 | 99.91 | - | 89.00 | 43.56 | 76.03 | **93.32** | 41.59 | 79.16 | 91.84 | **1.69** | **98.39** |
| | WaNet | 93.41 | 99.59 | - | 92.43 | 36.99 | 80.81 | **93.37** | 20.22 | 89.66 | 92.39 | **0.52** | **99.02** |
| | IAB | 93.57 | 98.81 | - | 93.30 | **0.32** | **99.11** | 93.40 | 1.54 | 98.55 | 92.37 | 0.38 | 98.62 |
| | Lira | 94.29 | 99.98 | - | 90.28 | 87.60 | 54.18 | **93.84** | 26.02 | 86.75 | 92.80 | **0.11** | **99.19** |
| | Average | 93.73 | 99.71 | - | 90.69 | 46.02 | 75.33 | **93.52** | 47.87 | 75.82 | 92.24 | **2.43** | **97.90** |
| GTSRB | BadNets | 95.08 | 100.00 | - | 94.51 | **0.30** | **99.56** | 95.08 | 98.81 | 50.59 | 94.27 | 6.78 | 96.20 |
| | Trojan | 94.39 | 100.00 | - | 93.62 | 45.74 | 76.75 | **94.61** | 99.20 | 50.40 | 93.40 | **0.49** | **99.27** |
| | Blend | 93.85 | 99.50 | - | 93.70 | 56.08 | 71.64 | **94.71** | 81.07 | 59.22 | 93.39 | **7.06** | **95.99** |
| | WaNet | 93.99 | 97.07 | - | 95.05 | 69.98 | 63.54 | **95.67** | 47.00 | 75.04 | 95.60 | **0.00** | **98.54** |
| | IAB | 94.09 | 97.22 | - | 93.17 | 31.11 | 82.59 | **94.42** | 11.85 | 92.68 | 94.21 | **0.00** | **98.61** |
| | Lira | 93.97 | 99.91 | - | 93.43 | 6.49 | 96.44 | **94.13** | 14.04 | 92.94 | 92.79 | **0.00** | **99.37** |
| | Average | 94.23 | 98.95 | - | 93.91 | 34.95 | 81.75 | **94.77** | 58.66 | 70.14 | 93.94 | **2.39** | **98.00** |
| TinyImageNet | BadNets | 61.98 | 99.97 | - | **59.38** | 0.00 | **98.68** | 58.03 | 0.34 | 97.84 | 56.33 | **0.00** | 97.16 |
| | Trojan | 61.58 | 100.00 | - | **58.05** | 0.06 | **98.20** | 56.95 | 0.16 | 97.60 | 56.71 | **0.01** | 97.56 |
| | Blend | 62.28 | 99.97 | - | **59.55** | 32.61 | 82.31 | 59.00 | 0.11 | **98.29** | 57.97 | **0.01** | 97.82 |
| | WaNet | 62.37 | 99.58 | - | 58.15 | **0.00** | 97.68 | 60.78 | 0.22 | 98.88 | **61.37** | 0.02 | **99.28** |
| | IAB | 62.56 | 99.39 | - | 59.20 | **0.00** | 98.01 | 60.40 | 0.03 | **98.60** | **61.12** | 2.39 | 97.78 |
| | Lira | 62.19 | 99.99 | - | 56.15 | 68.37 | 62.79 | 58.45 | 0.15 | 98.05 | **59.78** | **0.02** | **98.78** |
| | Average | 62.16 | 99.82 | - | 58.41 | 16.84 | 89.61 | **58.94** | **0.17** | **98.21** | 58.88 | 0.41 | 98.06 |

Table 8: Backdoor removal results of our proposed method for each size of the reference dataset.

| | | No Defense | | | Ours (1.0%) | | | Ours (5.0%) | | | Ours (10.0%) | | |
|---|---|---|---|---|---|---|---|---|---|---|---|---|---|
| | | ACC↑ | ASR↓ | DER↑ | ACC↑ | ASR↓ | DER↑ | ACC↑ | ASR↓ | DER↑ | ACC↑ | ASR↓ | DER↑ |
| CIFAR-10 | BadNets | 93.81 | 100.00 | - | 91.55 | 3.53 | 97.10 | 92.03 | 10.88 | 93.67 | 92.57 | 14.18 | 92.29 |
| | Trojan | 94.00 | 100.00 | - | 91.09 | 1.13 | 97.98 | 92.01 | 0.98 | 98.52 | 92.57 | 1.30 | 98.64 |
| | Blend | 93.29 | 99.91 | - | 90.61 | 0.52 | 98.35 | 91.84 | 1.69 | 98.39 | 92.11 | 1.08 | 98.83 |
| | WaNet | 93.41 | 99.59 | - | 89.38 | 0.19 | 97.69 | 92.39 | 0.52 | 99.02 | 92.61 | 0.37 | 99.21 |
| | IAB | 93.57 | 98.81 | - | 90.06 | 0.23 | 97.53 | 92.37 | 0.38 | 98.62 | 92.60 | 0.66 | 98.59 |
| | Lira | 94.29 | 99.98 | - | 91.15 | 0.16 | 98.34 | 92.80 | 0.11 | 99.19 | 92.83 | 0.11 | 99.20 |
| | Average | 93.73 | 99.71 | - | 90.64 | 0.96 | 97.83 | 92.24 | 2.43 | 97.90 | 92.55 | 2.95 | 97.79 |
| GTSRB | BadNets | 95.08 | 100.00 | - | 71.01 | 0.00 | 87.97 | 94.27 | 6.78 | 96.20 | 94.71 | 1.88 | 98.88 |
| | Trojan | 94.39 | 100.00 | - | 63.97 | 0.00 | 84.79 | 93.40 | 0.49 | 99.27 | 93.82 | 5.26 | 97.09 |
| | Blend | 93.85 | 99.50 | - | 68.18 | 0.00 | 86.91 | 93.39 | 7.06 | 95.99 | 94.24 | 2.82 | 98.34 |
| | WaNet | 93.99 | 97.07 | - | 80.58 | 0.00 | 91.83 | 95.60 | 0.00 | 98.54 | 95.44 | 0.00 | 98.54 |
| | IAB | 94.09 | 97.22 | - | 72.51 | 0.00 | 87.82 | 94.21 | 0.00 | 98.61 | 94.44 | 0.02 | 98.60 |
| | Lira | 93.97 | 99.91 | - | 64.78 | 0.00 | 85.36 | 92.79 | 0.00 | 99.37 | 93.40 | 0.00 | 99.67 |
| | Average | 94.23 | 98.95 | - | 70.17 | 0.00 | 87.45 | 93.94 | 2.39 | 98.00 | 94.34 | 1.66 | 98.52 |
| TinyImageNet | BadNets | 61.98 | 99.97 | - | 55.96 | 0.02 | 96.96 | 56.33 | 0.00 | 97.16 | 53.84 | 0.04 | 95.89 |
| | Trojan | 61.58 | 100.00 | - | 56.11 | 0.01 | 97.26 | 56.71 | 0.01 | 97.56 | 54.25 | 0.07 | 96.30 |
| | Blend | 62.28 | 99.97 | - | 55.12 | 0.00 | 96.40 | 57.97 | 0.01 | 97.82 | 53.62 | 0.01 | 95.65 |
| | WaNet | 62.37 | 99.58 | - | 57.67 | 0.00 | 97.44 | 61.37 | 0.02 | 99.28 | 56.87 | 0.03 | 97.02 |
| | IAB | 62.56 | 99.39 | - | 58.18 | 0.00 | 97.50 | 61.12 | 2.39 | 97.78 | 55.94 | 0.04 | 96.36 |
| | Lira | 62.19 | 99.99 | - | 57.34 | 0.00 | 97.57 | 59.78 | 0.02 | 98.78 | 55.98 | 0.07 | 96.85 |
| | Average | 62.16 | 99.82 | - | 56.73 | 0.01 | 97.19 | 58.88 | 0.41 | 98.06 | 55.08 | 0.04 | 96.35 |

## F.5 RESULTS FOR DIFFERENT REFERENCE DATASET SIZES

To investigate the dependency of our method on the size of the reference dataset, we further conducted experiments by varying the reference set at 1.0%, 5.0% and 10.0% of the training dataset. As shown in Table 8, our method consistently reduces ASR to nearly 0.0% across all dataset sizes, demonstrating that even a small reference set can effectively eliminate backdoors. Regarding clean accuracy, we observe that using 5.0% of the reference dataset already provides stable performance that is almost identical to using 10.0%, indicating that 5.0% is sufficient in practice.

However, we note that on GTSRB, using only 1.0% of the reference dataset significantly decreases accuracy (from 94.23% to 70.17%), although ASR is still effectively reduced to 0.0%. This result suggests that for datasets with complex distributions such as GTSRB, a slightly larger reference dataset (e.g., $\geq$ 5.0%) is required to preserve clean accuracy while maintaining strong defense efficacy.

## F.6 EFFECTIVENESS OF HYPERPARAMETER $\beta$

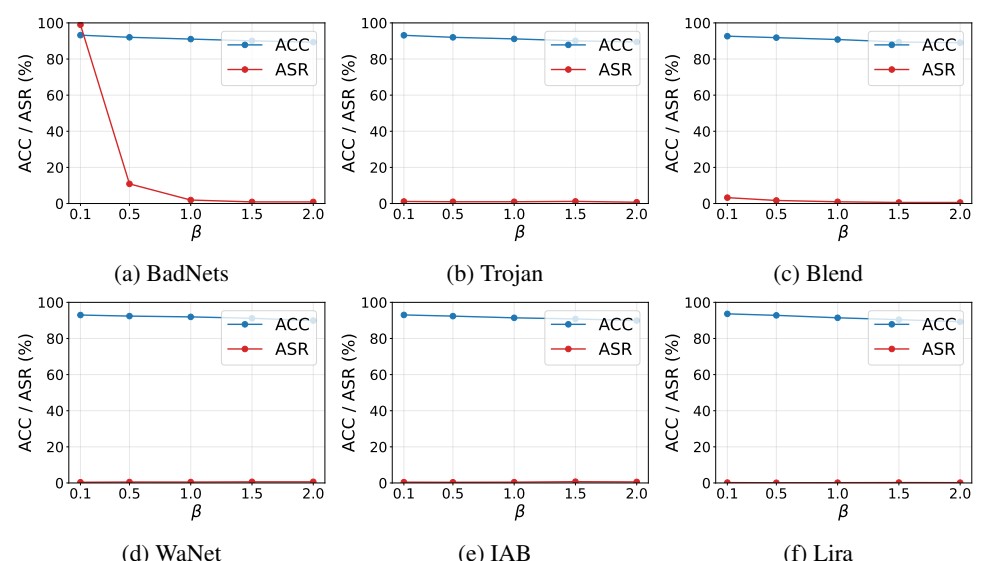

Figure 7: Effectiveness of the hyperparameter $\beta$ for CIFAR-10.

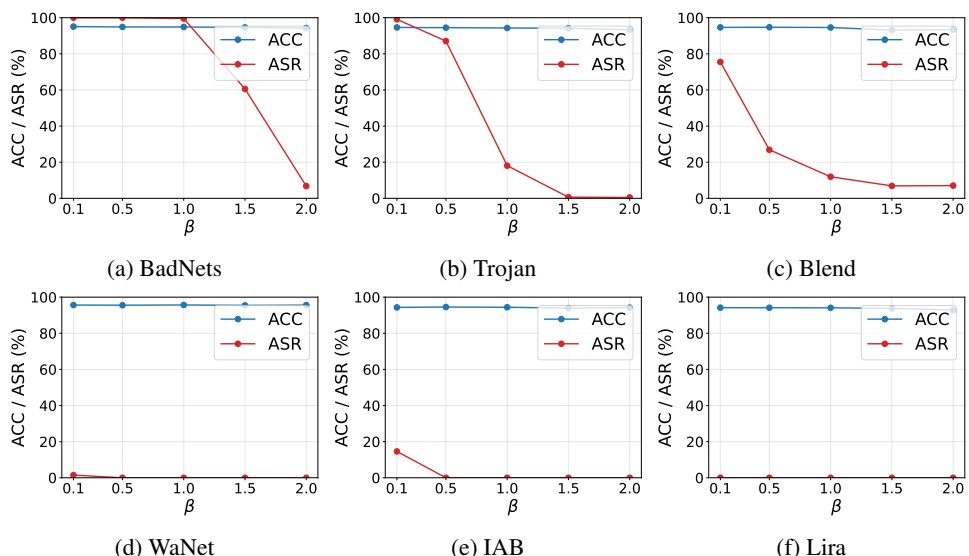

Figure 8: Effectiveness of the hyperparameter $\beta$ for GTSRB.

Since the hyperparameter $\beta$ is a crucial parameter that balances ACC and ASR, the parameter tuning for $\beta$ is very important process. Here, we present the tuning results obtained when $\beta$ is tuned manually and when they is tuned automatically.

**Manual Tuning**. Figure 7, Figure 8, and Figure 9 show how ACC and ASR vary with different values of $\beta$ for each dataset. We observe that as $\beta$ increases, both ACC and ASR decrease for all datasets. For CIFAR-10, we set $\beta = 0.5$ as it provides a good trade-off between ACC and ASR. For GTSRB, the ASR does not decrease unless $\beta$ is set to $2.0$ in some cases (e.g., BadNets and Blend). However, since the clean accuracy does not drop significantly, we set $\beta = 2.0$. For TinyImageNet, while ACC decreases substantially as $\beta$ increases, the ASR is reduced to nearly zero already at $\beta = 0.1$, and thus we set $\beta = 0.1$.

**Automatic Tuning**. To imshow the usability and robustness of our proposed method, we present the simple automatic tuning method where $\beta$ can be treated as a learnable scalar parameter and optimized jointly with the model parameters during the fine-tuning stage. Specifically, we initialize

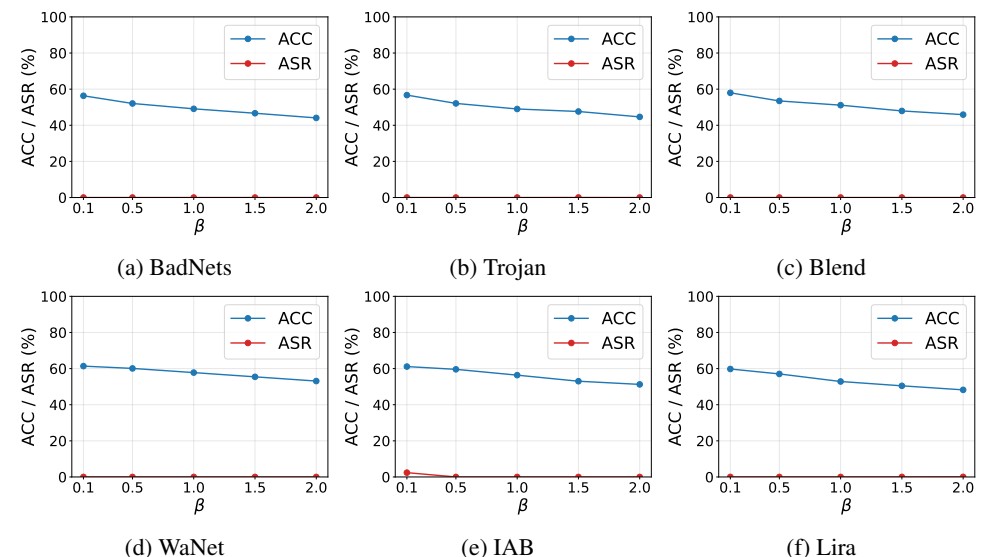

Figure 9: Effectiveness of the hyperparameter $\beta$ for TinyImageNet.

Table 9: Backdoor removal results for CIFAR-10 on ResNet-18 by manual and automatic tuning.

| | | No Defense | | | Ours (Manual) | | | Ours (Auto) | | |
|---|---|---|---|---|---|---|---|---|---|---|
| | | ACC↑ | ASR↓ | DER↑ | ACC↑ | ASR↓ | DER↑ | ACC↑ | ASR↓ | DER↑ |
| CIFAR-10 | BadNets | 93.81 | 100.00 | - | 92.03 | 10.88 | 93.67 | 91.28 | 1.34 | 98.27 |
| | Trojan | 94.00 | 100.00 | - | 92.01 | 0.98 | 98.52 | 91.41 | 1.29 | 98.06 |
| | Blend | 93.29 | 99.91 | - | 91.84 | 1.69 | 98.39 | 91.13 | 0.87 | 98.44 |
| | WaNet | 93.41 | 99.59 | - | 92.39 | 0.52 | 99.02 | 91.88 | 0.69 | 98.69 |
| | IAB | 93.57 | 98.81 | - | 92.37 | 0.38 | 98.62 | 91.83 | 0.46 | 98.31 |
| | Lira | 94.29 | 99.98 | - | 92.80 | 0.11 | 99.19 | 91.82 | 0.19 | 98.66 |
| | Average | 93.73 | 99.72 | - | 92.24 | 2.09 | 98.57 | 91.55 | 0.81 | 98.41 |
| GTSRB | BadNets | 95.08 | 100.00 | - | 94.27 | 6.78 | 96.20 | 94.89 | 47.03 | 76.39 |
| | Trojan | 94.39 | 100.00 | - | 93.40 | 0.49 | 99.27 | 93.84 | 0.00 | 99.73 |
| | Blend | 93.85 | 99.50 | - | 93.39 | 7.06 | 95.99 | 94.70 | 3.40 | 98.05 |
| | WaNet | 93.99 | 97.07 | - | 95.60 | 0.00 | 98.54 | 95.66 | 0.00 | 98.54 |
| | IAB | 94.09 | 97.22 | - | 94.21 | 0.00 | 98.61 | 94.37 | 0.00 | 98.61 |
| | Lira | 93.97 | 99.91 | - | 92.79 | 0.00 | 99.37 | 94.32 | 0.00 | 99.96 |
| | Average | 94.23 | 99.62 | - | 93.94 | 2.39 | 98.33 | 94.63 | 8.08 | 95.88 |
| TinyImageNet | BadNets | 61.98 | 99.97 | - | 56.33 | 0.00 | 97.16 | 50.35 | 0.00 | 94.17 |
| | Trojan | 61.58 | 100.00 | - | 56.71 | 0.01 | 97.56 | 51.38 | 0.00 | 94.90 |
| | Blend | 62.28 | 99.97 | - | 57.97 | 0.01 | 97.82 | 52.29 | 0.01 | 94.98 |
| | WaNet | 62.37 | 99.57 | - | 61.37 | 0.02 | 99.28 | 58.66 | 0.02 | 97.92 |
| | IAB | 62.56 | 99.39 | - | 61.12 | 2.39 | 97.78 | 57.54 | 0.00 | 97.18 |
| | Lira | 62.19 | 99.99 | - | 59.78 | 0.02 | 98.78 | 55.22 | 0.02 | 96.50 |
| | Average | 62.16 | 99.82 | - | 58.55 | 0.41 | 97.89 | 54.91 | 0.01 | 95.74 |

$\beta$ with a reasonable default value and update it through gradient-based optimization along with the model weights. In Table 9, we show the experimental results for CIFAR-10, GTSRB, and TinyImageNet on ResNet-18. We use the initial $\beta = \log(1 + e^u)$ with $u = 3.0$, so that a softplus function is applied to prevent $\beta$ from taking negative values. Here, we directly update $u$, which serves as an input to the softplus function. For manual tuning, we follow the original experimental settings and set the values to 0.5, 2.0, and 0.1 for CIFAR-10, GTSRB, and TinyImageNet, respectively.

Although the initial value of $\beta$ or $u$ must be specified, our experimental results indicate $\beta$ rapidly converges to a stable range regardless of its initialization and that the resulting performance is comparable to that obtained with manually tuned $\beta$. Here, we confirmed that each final value of $\beta$ does not collapse to 0 and remains meaningful. It is possible that, during training, the loss term multiplied by $\beta$ became quite small and eventually received almost no updates. In summary, our results of the

Table 10: Our poisoned class identification results for low poisoning rates (PR) for CIFAR-10 on ResNet-18.

| | PR = 1.0% | | | PR = 5.0% | | |
|---|---|---|---|---|---|---|
| | $p_{\text{val}}$ | $N^*$ | Poisoned Class | $p_{\text{val}}$ | $N^*$ | Poisoned Class |
| Clean | 0.998 | 14 | - | 0.998 | 14 | - |
| BadNets | 0.0000226 | 30 | 1 | 0.000312 | 29 | 1 |
| Trojan | 0.7304 | 24 | - | 0.00211 | 28 | 1 |
| Blend | 0.0000226 | 30 | 1 | 0.0000226 | 30 | 1 |

Table 11: Backdoor removal results under low poisoning rates for CIFAR-10 on ResNet-18.

| | No Defense | | | Ours | | |
|---|---|---|---|---|---|---|
| | ACC↑ | ASR↓ | DER↑ | ACC↑ | ASR↓ | DER↑ |
| BadNets (PR = 1.0%) | 93.92 | 100.00 | - | 91.69 | 1.48 | 98.15 |
| BadNets (PR = 5.0%) | 93.91 | 100.00 | - | 91.62 | 20.27 | 88.72 |
| Trojan (PR = 1.0%) | 93.96 | 99.97 | - | 91.62 | 1.07 | 98.28 |
| Trojan (PR = 5.0%) | 93.78 | 100.00 | - | 90.92 | 1.30 | 97.92 |
| Blend (PR = 1.0%) | 93.91 | 97.53 | - | 91.50 | 1.00 | 97.06 |
| Blend (PR = 5.0%) | 93.67 | 99.78 | - | 91.00 | 0.58 | 98.27 |

simple automatic tuning indicate that a large portion of the tuning effort can be automated without requiring dataset- or model-specific heuristics.

## F.7 EFFECTIVENESS FOR LOW POISONING RATES

To exhibit the validity of our proposed method under low poisoning rates because when the poisoning rate becomes very small, the influence of the backdoor on the latent representation naturally diminishes, which may make it difficult to detect the poisoned class and our fine-tuning method may not work well.

First, we show our poisoned class identification results under low poisoning rates of 1.0% and 5.0% in Table 10.

For 5.0% poisoning rate, our method reliably identifies the poisoned class for all three attacks, with $p_{\text{val}} \approx 0.0$ and the minimal-norm class consistently selected $N^* \approx 30$. This shows our method remains robust at moderately low poisoning rates. At the 1.0% poisoning rate, Trojan yields a higher $p_{\text{val}} = 0.7304$, above the significance threshold, indicating its perturbation norms become indistinguishable from clean classes. This is expected, as Trojan triggers become more subtle and cause weaker shifts in the latent representation, and very low poisoning rates further reduce their effect on the compromised model.

We further show the backdoor removal results in Table 11. Our results indicate that the proposed method is generally effective even under low poisoning rates, the poisoned class is reliably detected for most of attacks, and the subsequent fine-tuning stage substantially suppresses the backdoor while preserving clean accuracy. Moreover, even in the rare cases where poisoned class identification becomes unreliable at extremely low poisoning rates (e.g., Trojan at PR = 1.0%), the fine-tuning step still remains highly effective, consistently reducing ASR. This indicates that the final defense stage is robust enough to mitigate backdoor behavior even when the identification component is challenged.

## F.8 EFFECTIVENESS AGAINST DEFENSE-AWARE ATTACK

An adaptive adversary with access to the model training process could deliberately enforce smaller trigger-activated changes (TAC) for the poisoned class. Such an adversary may attempt to directly minimize the $L^2$ norm of the perturbations associated with the poisoned class during training. A

Table 12: Backdoor removal results against the defense-aware attack on ResNet-18.

| | No Defense | | | Ours | | |
|---|---|---|---|---|---|---|
| | ACC↑ | ASR↓ | DER↑ | ACC↑ | ASR↓ | DER↑ |
| CIFAR-10 | 91.90 | 100.00 | - | 91.35 | 1.09 | 99.18 |
| GTSRB | 94.38 | 100.00 | - | 94.68 | 0.01 | 100.00 |
| TinyImageNet | 41.32 | 99.34 | - | 60.22 | 0.15 | 99.59 |

representative defense-aware objective can be formulated as follows:

$$\boldsymbol{\theta}_{\text{bd}} = \operatorname*{argmin}_{\boldsymbol{\theta}} \frac{1}{n} \sum_{i=1}^{n} \left[ \ell(f(\boldsymbol{x}_i; \boldsymbol{\theta}), \boldsymbol{y}_i) + \ell(f(\boldsymbol{x}_i + \boldsymbol{\delta}; \boldsymbol{\theta}), \boldsymbol{e}_t) + \lambda \underbrace{\|\phi_{\boldsymbol{\theta}}(\boldsymbol{x} + \boldsymbol{\delta}) - \phi_{\boldsymbol{\theta}}(\boldsymbol{x})\|_2}_{\text{minimize TAC}} \right]. \quad (8)$$

This loss explicitly encourages the adversary to train a model whose latent representations are close between clean data and poisoned data, thereby minimizing the TAC in the latent representation.

To verify the effectiveness of our proposed method against the defense-aware attack, we conducted defense experiments agaisnt the compromised model which is trained by the objective as shown in Equation (8) using the BadNets trigger.

First, when we tested the poisoned class identification method, we confirmed that the poisoned class was correctly identified in all datasets, showing $p_{\text{val}} = 0.000023$ and $N^* = 30$. Furthermore, the backdoor removal results for CIFAR-10, GTSRB, and TinyImageNet are also shown in Table 12.

Across all datasets, we find that our defense remains highly effective, reducing ASR to 1.09%, 0.01%, and 0.15%, respectively, while maintaining clean accuracy almost unchanged and increased for GTSRB and TinyImageNet. This is because adding the term that minimizes TAC degraded the clean accuracy of the compromised model, and our fine-tuning subsequently restored the accuracy.

These results demonstrate that even when the adversary explicitly attempts to minimize TAC in the latent representation, our defense continues to correctly identify the poisoned class and successfully suppress the backdoor behavior.

### F.9 VISUALIZATION FOR RECONSTRUCTED TAC

We show the reconstructed and true TAC values in Figure 10. From the visualization, we can see that the reconstructed values are very close to the true TAC and are reconstructed accurately.

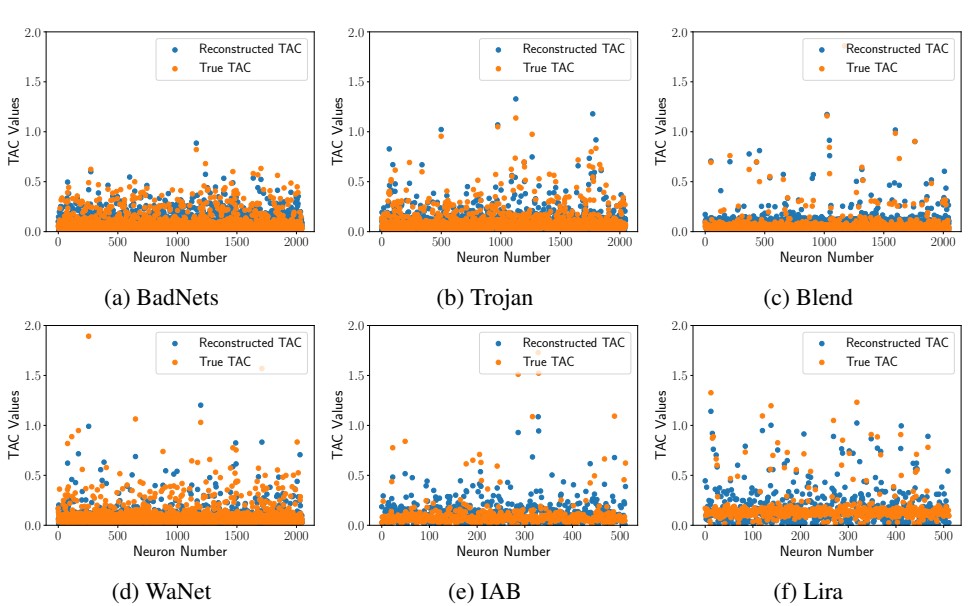

Figure 10: Visualization for Reconstructed TAC for CIFAR-10 on ResNet-50.

