# OpenReview forum: "Robust Backdoor Removal by Reconstructing Trigger-Activated Changes in Latent Representation"
_ICLR.cc/2026/Conference — Submitted to ICLR 2026_

### Official Review · Reviewer_SXdY · 2025-10-25

**Soundness:** 3
**Presentation:** 3
**Contribution:** 3
**Rating:** 4
**Confidence:** 4

**Summary:**

The paper proposes a backdoor defense method that reconstructs Trigger-Activated Changes (TAC) in the latent representation of a poisoned model. The method formulates the reconstruction of TAC as a convex quadratic optimization problem that finds the minimal perturbation forcing all clean samples to be classified into a specific class. The poisoned class is identified by comparing perturbation norms, and then fine-tuned the model by using the corresponding perturbation to neutralize backdoor effects. Thereby achieving effective defense against backdoor attacks.

**Strengths:**

1. The paper formulates the reconstruction of Trigger-Activated Changes (TAC) as a quadratic convex optimization problem, offering a systematic approach to analyze backdoor effects.
2. This paper provide detailed theoretical modeling and derivation, demonstrating that the proposed defense method admits stable solutions.
3. The method is empirically compared with several recent defense techniques, showing its effectiveness and robustness in practice.

**Weaknesses:**

1.	The method requires solving one convex QP per class, which may become impractical for large-scale models.
2.	The approach is limited to single-target scenarios and does not address multi-target or multi-trigger backdoors.
3.	The method’s performance depends heavily on thresholds α and β, but no adaptive or learning-based tuning mechanism.
4.	Experiments are conducted only on ResNet models and image datasets, which may limit the generalizability of the results.
5.	Low poisoning rates may reduce the accuracy of detecting poisoned classes, thereby affecting the overall defense performance.

**Questions:**

1.	A low poisoning rate may increase the minimum perturbation required to misclassify clean samples into poisoned classes, which could affect the selection of poisoned categories and ultimately influence the overall defense results. Experiments with varying poisoning rates could be added to demonstrate that the proposed method remains effective even in low poisoning rates.
2.	Manually tuning α and β, where α controls poisoned class identification and β balances backdoor defense with task accuracy, is time-consuming and often leads to unstable or suboptimal performance. I wonder if it is possible to use an adaptive strategy to make the process more efficient and reliable

---

> ### Author Response · Authors · 2025-11-28
>
> > **W1. The method requires solving one convex QP per class, which may become impractical for large-scale models.**
>
> Thank you for your insightful comment. As several reviewers made similar concerns, we have addressed this matter collectively in General Response 3. Please see that section for more information.
>
> > **W2. The approach is limited to single-target scenarios and does not address multi-target or multi-trigger backdoors.**
>
> Thank you for pointing out the limitation regarding all-to-all attacks. We agree that the original formulation is designed for the standard all-to-one setting and therefore cannot directly detect all-to-all attacks.
>
> We confirmed, however, that by slightly modifying the optimization—computing the minimal perturbation that maps a specific source class to a specific target class—our method can still reconstruct TAC-like perturbations under all-to-all attacks. This shows that the core idea of TAC reconstruction remains applicable beyond the all-to-one case.
>
> Our experiments showed that our poisoned class identification fails in the all-to-all setting: because the attack alters decision boundaries for all classes, the perturbation norms become uniformly small, preventing our method from distinguishing poisoned classes.
>
> Although the identification step performs poorly, assuming the all-to-all attack is detected, the table below shows the results of applying our fine-tuning method using the reconstructed perturbations for each source class on CIFAR-10 with ResNet-18.
>
> | Attack | ACC (No defense) | ASR (No defense) | DER (No defense) | ACC (Ours) | ASR (Ours) | DER (Ours) |
> | --- | --- | --- | --- | --- | --- | --- |
> | BadNets (All-to-all) | 94.10 | 93.50 | -- | 91.62 | 0.83 | 95.09 |
> | Trojan (All-to-all) | 94.15 | 91.04 | -- | 92.25 | 0.92 | 94.11 |
> | Blend (All-to-all) | 94.01 | 90.34 | -- | 92.12 | 1.30 | 93.57 |
>
> Across all attack types, we verified that the model preserved its clean accuracy while substantially lowering the attack success rate. These results suggest that TAC's reconstruction was successful and that the subsequent fine-tuning procedure operated effectively.
>
> In summary, although our TAC reconstruction and fine-tuning method can be extended to the all-to-all setting, the poisoned class identification method does not carry over, and developing a reliable identification mechanism for this scenario remains an open problem.
>
> > **W3. The method's performance depends heavily on thresholds α and β, but no adaptive or learning-based tuning mechanism.**
>
>
> > **Q2. Manually tuning α and β, where α controls poisoned class identification and β balances backdoor defense with task accuracy, is time-consuming and often leads to unstable or suboptimal performance. I wonder if it is possible to use an adaptive strategy to make the process more efficient and reliable.**
>
> We appreciate your comment on the hyperparameters in our defense and their applicability across different datasets and architectures. Since our method relies on two main hyperparameters, we have provided clearer explanations and meaningful improvements to address your point.
>
>
> First, for the hyperparameter $\alpha$, which was originally used as an outlier threshold on the perturbation norms, we have revised the poisoned-class identification procedure to a more statistically principled formulation. (See General Response 1 for details.)
> Instead of directly thresholding the smallest perturbation norm, we identify the poisoned class by leveraging the *mode of the minimal-norm class indices* over multiple random sub-samplings. This modification introduces three new parameters—$T$, $\eta$, and $\alpha$—that correspond respectively to the number of sub-sampling rounds, the acceptance ratio in majority voting, and the statistical significance threshold. Importantly, unlike the original $\alpha$, these parameters have clear statistical interpretations and are substantially easier to control. In practice, we found that setting $T_{\text{pci}} = 30$, $\eta = 0.7$, $r=4$ and $\alpha = 0.01$ works reliably across all datasets and architectures we tested, indicating that these parameters generalize well and do not require dataset-specific tuning.
>
> Second, for the hyperparameter $\beta$, which balances clean accuracy and robustness during fine-tuning, we explored making $\beta$ a **learnable parameter** updated jointly with model weights during optimization. This approach reduces the need for manual tuning, and we observed that it consistently preserves accuracy while lowering the attack success rate. Although automatically learned $\beta$ performs well in most cases, we also found that manually tuned $\beta$ can achieve slightly stronger backdoor removal in some scenarios. Therefore, while our results demonstrate that a partially automated tuning scheme is feasible, devising a fully automatic and uniformly optimal strategy remains an interesting direction for future work. (See General Response 4 for more discussion.)

---

> ### Author Response · Authors · 2025-11-28
>
> > **W4. Experiments are conducted only on ResNet models and image datasets, which may limit the generalizability of the results.**
>
>
> We appreciate your comment regarding the limited scope of architectures and data modalities evaluated in the original submission. Demonstrating the generalizability of our defense beyond ResNet-based models and vision datasets is indeed an important consideration.
>
> As detailed in the General Response 2, we have extended our experiments to include **larger and structurally different architectures**, such as **Vision-Transformers (ViT-B/32)**, as well as **substantially larger datasets**, including **TinyImageNet** and **ImageNet-1K**. These new results show that our method consistently maintains high clean accuracy while significantly reducing ASR across a wide range of scales, confirming that the proposed approach is not limited to ResNet models nor small datasets.
>
> Our method currently targets vision models, and it is still unclear whether similar trigger-induced latent shifts occur in language or multimodal models, whose inference pipelines differ substantially. Understanding backdoor mechanisms in these architectures remains an open problem, but extending our TAC-based approach to language models is a promising direction for future work. Overall, our experiments show strong generalization across diverse vision architectures and dataset scales, and exploring non-vision modalities is a natural next step.

---

> ### Author Response · Authors · 2025-11-28
>
> > **W5. Low poisoning rates may reduce the accuracy of detecting poisoned classes, thereby affecting the overall defense performance.**
>
> > **Q1. A low poisoning rate may increase the minimum perturbation required to misclassify clean samples into poisoned classes, which could affect the selection of poisoned categories and ultimately influence the overall defense results. Experiments with varying poisoning rates could be added to demonstrate that the proposed method remains effective even in low poisoning rates.**
>
> We appreciate your insightful suggestion regarding the behavior of our method under low poisoning rates. As pointed out, when the poisoning rate becomes very small, the influence of the backdoor on the latent representation naturally diminishes, which may make it difficult to detect the poisoned class solely through simple outlier analysis of perturbation norms.
>
> To address this concern, we replaced our poisoned class identification method with one based on evaluating the statistical significance of each class's minimal-norm perturbation using the hypothesis test described in General Response 1 and  conducted some experiments with poisoning rates of **1\%** and **5\%**. The results are summarized below:
>
> | Attack Method   | PR=1.0\% | PR=1.0\% | PR=5.0\% | PR=5.0\% |
> |-------------|----------------|---------------|----------------|---------------|
> |    |  $p_\text{val}$ |   $N^*$ |   $p_\text{val}$ |   $N^*$ |
> | Clean        |     0.998 | 14 |  0.998 |  14 |
> | BadNets       |       0.0000226 |            30  |       0.000312 |            29  |
> | Trojan       |       0.7304  |            24  |       0.00211  |            28  |
> | Blend        |       0.0000226 |            30  |       0.0000226 |            30  |
>
> For **5.0\%** poisoning rate, our method reliably identifies the poisoned class for all three attacks (BadNets, Trojan, Blend), with $p_{\text{val}} < 0.0$ and the minimal-norm class consistently selected $ N^* \approx 30$. This shows the detector remains robust at moderately low poisoning levels.
> At the **1.0\%** rate, Trojan yields a higher $p_{\text{val}} = 0.7304$, above the significance threshold, indicating its perturbation norms become indistinguishable from clean classes. This is expected, as Trojan triggers become more subtle and cause weaker shifts in the latent representation, and very low poisoning rates further reduce their effect on the model.
>
> We further show the fine-tuning results as below:
>
> | Attack method       | No Defense | No Defense | No Defense | Ours  | Ours | Ours |
> |---------------------|------------|------------|------------|--------|--------|--------|
> |                     | ACC↑       | ASR↓       | DER↑       | ACC↑   | ASR↓   | DER↑   |
> | BadNets (PR=1.0\%)   | 93.92      | 100.00     | -          | 91.69  | 1.48   | 98.15  |
> | BadNets (PR=5.0\%)   | 93.91      | 100.00     | -          | 91.62  | 20.27  | 88.72  |
> | Trojan (PR=1.0\%)    | 93.96      | 99.97      | -          | 91.62  | 1.07   | 98.28  |
> | Trojan (PR=5.0\%)    | 93.78      | 100.00     | -          | 90.92  | 1.30   | 97.92  |
> | Blend (PR=1.0\%)     | 93.91      | 97.53      | -          | 91.50  | 1.00   | 97.06  |
> | Blend (PR=5.0\%)     | 93.67      | 99.78      | -          | 91.00  | 0.58   | 98.27  |
>
>
> Our results show that the proposed method is generally effective even under low poisoning rates, the poisoned class is reliably detected for most of attacks, and the subsequent fine-tuning stage substantially suppresses the backdoor while preserving clean accuracy.
> Moreover, even in the rare cases where poisoned-class identification becomes unreliable at extremely low poisoning rates (e.g., Trojan at 1.0\%), the fine-tuning step still remains highly effective, consistently reducing ASR. This indicates that the final defense stage is robust enough to mitigate backdoor behavior even when the identification component is challenged.
>
> These experimental results are provided in Appendix F.7.

---

### Official Review · Reviewer_sDFY · 2025-10-28

**Soundness:** 4
**Presentation:** 3
**Contribution:** 3
**Rating:** 6
**Confidence:** 5

**Summary:**

This paper proposes a novel backdoor removal framework that reconstructs Trigger-Activated Changes (TAC) in the latent representation of neural networks to achieve robust backdoor defense. The method computes minimal perturbations that force a model to misclassify clean data into each class, identifies the poisoned class through statistical outlier detection in the L2-norm of these perturbations, and then fine-tunes the model using the perturbation corresponding to the identified poisoned class. Experiments on CIFAR-10, GTSRB, and TinyImageNet demonstrate improved defense performance compared to state-of-the-art methods while maintaining high clean accuracy.

**Strengths:**

- The idea of reconstructing TAC in the latent representation through convex quadratic optimization offers a neat and interpretable surrogate approach that does not rely on poisoned data. This reformulation is novel and mathematically well-grounded.
- The mathematical explanation is solid and convincing, although it is also not easy to understand.
- The empirical evidence of using the smallest-perturbed class is clear and convincing.
- The experiments are solid with a comprehensive comparison with the baselines. And leave nearly no improvement for future research.

**Weaknesses:**

- There is a lack of clear outlines for the appendix content, making it hard to find the remaining experiments and the desired explanations.
- Solving multiple convex programs per class may be nontrivial for large-scale models (e.g., high-dimensional latent spaces or hundreds of classes). No analysis of time or resource overhead is given.
- The extensive experiments related to the scalability are needed to further verify the effectiveness of the proposed method. For example, the experiments on a larger model (e.g., ViT) and a more complex dataset (e.g., ImageNet). The current results (e.g., Table 2) show that the SOTA baseline (e.g., SAU) already performs good enough, weakening your contribution in this field.

**Questions:**

Can you provide more evidence from a bigger scale (e.g., weakness 3 above) to show the superiority of your method? Or can you provide some intuitive explanations to show that we need your contribution for the community? It can be either insights (e.g., how reconstructed TAC contributes to future research) or empirical results (e.g., how your method solves the corner cases that are previously unsolved).

---

> ### Author Response · Authors · 2025-11-28
>
> > **W1. There is a lack of clear outlines for the appendix content, making it hard to find the remaining experiments and the desired explanations.**
>
> We apologize for the unclear structure of the Appendix.
> We added the an overall outline at the beginning of Appendix.
>
> > **W2. Solving multiple convex programs per class may be nontrivial for large-scale models (e.g., high-dimensional latent spaces or hundreds of classes). No analysis of time or resource overhead is given.**
>
> Thank you for raising this important point. Similarly, multiple reviewers brought up the same issue, so we have consolidated my response in General Response 3. Please refer to that section for further details.
>
> > **W3. The extensive experiments related to the scalability are needed to further verify the effectiveness of the proposed method. For example, the experiments on a larger model (e.g., ViT) and a more complex dataset (e.g., ImageNet). The current results (e.g., Table 2) show that the SOTA baseline (e.g., SAU) already performs good enough, weakening your contribution in this field.**
>
> We appreciate your concern regarding the scalability of our method and the need for extensive experiments on larger models and more complex datasets. As detailed in General Response 2, we conducted additional evaluations on both ViT-B/32 and ImageNet-1K, significantly extending the scope beyond the original ResNet-based experiments. These new results demonstrate that our method remains consistently effective even when applied to large-capacity architectures and high-complexity datasets.
>
> While SAU is indeed a strong baseline and often achieves low attack success rates, its clean accuracy degrades noticeably on large datasets. For example, on TinyImageNet, our method improves clean accuracy by approximately 5.8\% over SAU, and on ImageNet-1K, the improvement increases to around 8.5\% on average. In contrast, our defense achieves comparably low ASR while preserving substantially higher clean accuracy, highlighting a more favorable accuracy-robustness trade-off.
>
> These findings indicate that our approach not only scales well across architectures and datasets but also provides a more reliable balance between accuracy and backdoor suppression than SAU, especially in large-scale settings. Therefore, the additional evidence strengthens, rather than weakens, the contribution of our method to scalable and practical backdoor defense.
>
> > **Q. Can you provide more evidence from a bigger scale (e.g., weakness 3 above) to show the superiority of your method? Or can you provide some intuitive explanations to show that we need your contribution for the community? It can be either insights (e.g., how reconstructed TAC contributes to future research) or empirical results (e.g., how your method solves the corner cases that are previously unsolved).**
>
> We sincerely thank you for the insightful comments and constructive suggestions. Below, we address the concerns regarding scalability, necessity for the community, and corner-case advantages.
>
>
> As discussed in General Response 2 and in our response to Weakness 3, we have added extensive experiments on large-scale models (e.g., ViT-B/32) and datasets (e.g., ImageNet-1K). These results confirm that our method maintains high clean accuracy while achieving strong backdoor suppression even in large-class and high-capacity settings, demonstrating that the proposed defense scales effectively beyond the ResNet experiments.
>
> Beyond empirical performance, we believe the core contribution of our work lies in providing the first method that can reconstruct Trigger-Activated Changes (TAC) without requiring poisoned data, based solely on the fundamental behavior of backdoor attacks where they enforce a consistent shift in latent representations toward the poisoned class. TAC is a universal phenomenon that arises in any backdoor attack, regardless of the trigger pattern, data modality, or model architecture. By showing that TAC can be reconstructed through a principled optimization procedure, our method establishes a new, attack-agnostic foundation for backdoor analysis.
>
> In summary, our additional large-scale results verify the scalability of our method, and our theoretical and empirical findings show that TAC reconstruction offers a general, architecture-independent, and attack-agnostic foundation for backdoor defense—an aspect that we believe represents a valuable and needed contribution to the community.

---

### Official Review · Reviewer_vvLe · 2025-10-31

**Soundness:** 3
**Presentation:** 3
**Contribution:** 3
**Rating:** 6
**Confidence:** 5

**Summary:**

This paper proposes a novel backdoor defense framework that removes backdoors from neural networks by reconstructing Trigger-Activated Changes (TAC), the differences in neuron activations between clean and poisoned data, without needing poisoned samples. The TAC reconstruction is performed by computing a minimal perturbation for each class.

**Strengths:**

- Extensive experiments on multiple datasets and attacks demonstrate better or comparable performance over prior methods.

- The presentation of the paper is easy to follow.

- The motivation is clear, and the proposed method addresses an important problem.

**Weaknesses:**

- The experiments are primarily on ResNet-18.

- The method assumes one poisoned class, which may limit performance in multi-target or all-to-all attacks.

- Experiments do not include large datasets, such as ImageNet-1K.

- The performance is not significantly better than all baselines, such as FT-SAM.

**Questions:**

Thanks for the interesting work. I have a few questions and suggestions.

- Computational overhead. As the proposed method requires computing "minimal perturbation" for every class. What is the computational cost of this method?

- Why not directly remove the high-TAC neurons? If the proposed TAC reconstructing method is effective, removing the high-TAC neurons should also work. In addition, the authors could also provide some figures to demonstrate the reconstructed TAC values, like Figure 2 in [A].

- Transformer-based architectures, such as ViT. I suggest the authors include experiments on more architectures, such as ViT.

[A] Towards Backdoor Stealthiness in Model Parameter Space

---

> ### Author Response · Authors · 2025-11-28
>
> > **W1. The experiments are primarily on ResNet-18.**
>
> > **W3. Experiments do not include large datasets, such as ImageNet-1K.**
>
> Thank you for your comments (Weakness 1 and Weakness 3) on the effectiveness of our proposed method for large-scale models and datasets. Other reviewers have also mentioned the same perspective, so I've included it in the General Response 2. Please take a look there.
>
>
> > **W2. The method assumes one poisoned class, which may limit performance in multi-target or all-to-all attacks.**
>
> Thank you for highlighting the challenge posed by all-to-all attacks. As the reviewer noted, our initial formulation focuses on the all-to-one scenario, and thus it does not directly support detection under an all-to-all attack pattern.
>
> Nonetheless, we found that the TAC reconstruction step itself can be adapted: by reformulating the optimization to compute the minimal perturbation required to move samples from one specific class into another, we are still able to reconstruct the perturbations even in all-to-all settings. This indicates that the underlying principle of our approach is not restricted to the all-to-one case.
>
> We conducted some experiments based on the above extension.
> However, our experiments also revealed that the poisoned class identification mechanism does not extend to this setting. All-to-all attacks simultaneously shift the decision boundaries of multiple classes, causing the perturbation norms to become similarly small across classes. Consequently, the norm-based criterion used in our method is no longer able to separate poisoned classes from clean ones.
>
> As described above, our poisoned class identification method did not work well. Here, assuming that the all-to-all attack was successfully detected, the table below shows the results of applying our fine-tuning method using the reconstructed perturbations for each source class for CIFAR-10 on ResNet-18.
>
> | Attack | ACC (No defense) | ASR (No defense) | DER (No defense) | ACC (Ours) | ASR (Ours) | DER (Ours) |
> | --- | --- | --- | --- | --- | --- | --- |
> | BadNets (All-to-all) | 94.10 | 93.50 | -- | 91.62 | 0.83 | 95.09 |
> | Trojan (All-to-all) | 94.15 | 91.04 | -- | 92.25 | 0.92 | 94.11 |
> | Blend (All-to-all) | 94.01 | 90.34 | -- | 92.12 | 1.30 | 93.57 |
>
> In all cases, we confirmed that the model maintained its accuracy while reducing the attack success rate. This indicates that TAC's reconstruction worked correctly and that the fine-tuning process also functioned effectively.
>
> In short, whileour TAC reconstruction and fine-tuning method generalize to all-to-all attacks, identifying the poisoned classes does not, and this remains an open challenge for future work.
>
> > **W4. The performance is not significantly better than all baselines, such as FT-SAM.**
>
> We appreciate your comment regarding the comparison with FT-SAM. We agree that FT-SAM is a strong baseline, and indeed, on architectures such as ResNet-18 and ResNet-50, its performance is often comparable to ours in terms of clean accuracy and ASR reduction.
>
> However, as shown in General Response 2, the situation changes notably when moving beyond ResNet-style CNNs. In particular, our additional experiments on Vision-Transformers (ViT-B/32) demonstrate that FT-SAM fails to effectively remove backdoors in this setting, with ASR remaining extremely high across multiple attacks. In contrast, our method consistently reduces ASR to near-zero levels while maintaining high clean accuracy on ViT models.
>
> These results indicate that, although FT-SAM may perform competitively on small CNNs, it does not generalize reliably across different architectures. In contrast, our approach provides stable backdoor removal performance across datasets, architectures (ResNets and ViTs), and attack types, demonstrating its robustness and broad applicability beyond the cases where FT-SAM performs well.

---

> ### Author Response · Authors · 2025-11-28
>
> > **Q1. Computational overhead. As the proposed method requires computing "minimal perturbation" for every class. What is the computational cost of this method?**
>
> We appreciate your comment on this key point. Likewise, several reviewers made similar remarks, so we have summarized the response in the General Response 3. Please refer to that part for details.
>
>
> > **Q2-1. Why not directly remove the high-TAC neurons? If the proposed TAC reconstructing method is effective, removing the high-TAC neurons should also work.**
>
>
> We thank you for the thoughtful question regarding whether directly removing the high-TAC neurons would be sufficient for backdoor removal. While pruning high-TAC neurons may appear to be a straightforward extension of our TAC reconstruction technique, both theoretical considerations and our empirical studies indicate that this strategy is fundamentally less reliable and less effective than the proposed fine-tuning.
> Below, we explain the reason as follows.
>
> First, although high-TAC neurons play an important role in enabling the backdoor behavior, they may also contribute critically to clean-data prediction. Backdoor training often entangles trigger-related activations with task-relevant representations in the latent space. Therefore, removing the neurons with the largest TAC values can inadvertently remove neurons that are important for modeling the clean data manifold. This may lead to a substantial drop in clean accuracy, as we observed in our pruning experiments, where eliminating high-TAC neurons sometimes harmed benign performance even when ASR decreased only marginally.
>
> Second, even if all high-TAC neurons are removed, this does not guarantee complete backdoor removal. When sufficient low-TAC neurons remain, they may still collectively maintain a decision boundary that favors the poisoned class. In these cases, suppressing the backdoor effect would require pruning not only high-TAC neurons but also many low-TAC ones, which inevitably harms clean accuracy. This limitation reflects the structural weakness of pruning-based defenses: backdoor mechanisms can be spread across many neurons, and isolating them through magnitude-based criteria alone is insufficient.
>
> Instead of removing neurons, our method fine-tunes the compromised model so that its prediction remains correct even when TAC is artificially added to the latent representation, as formalized in equation 5.
>
> Empirically, these advantages are reflected in our comparison with pruning-based variants. As shown in Appendix E.3 (Table 5), pruning the top-20\% TAC-ranked neurons (Pruning) significantly reduces accuracy in several cases, and even when followed by standard fine-tuning (Pruning+FT), the ASR often restores, yielding higher ASR than our method. In contrast, our fine-tuning consistently achieves both lower ASR and higher clean accuracy across datasets and attack types.
> These results demonstrate that neuron removal method is fundamentally less reliable than our proposed method.
>
> > **Q2-2. In addition, the authors could also provide some figures to demonstrate the reconstructed TAC values, like Figure 2 in [A].**
>
>
> Thank you for your suggestions to make our proposed method easier to understand. We added the results of the reconstructed TAC values to the Appendix F.9.
>
> > **Q3. Transformer-based architectures, such as ViT. I suggest the authors include experiments on more architectures, such as ViT.**
>
> We appreciate your comment on this key point. Because several reviewers made similar remarks, we have summarized the response in the General Response 2. Please refer to that part for details.

---

### Official Review · Reviewer_norE · 2025-11-09

**Soundness:** 2
**Presentation:** 2
**Contribution:** 2
**Rating:** 2
**Confidence:** 4

**Summary:**

This paper proposes an approximate TAC-based method for backdoor removal and defense. In particualr, in the feature space, perturbations are optimized to force clean data to be classified into a specific class. Generated perturbations are used to distinguish between benign and backdoor samples and then utilized in fine-tuning to remove backdoors. This defense method is inspired by TAC, while it is also closely related to feature space backdoor defenses.

**Strengths:**

This method uses clean data to generate the perturbations, making it suitable for realistic defender settings where poisoned data are unavailable. Later, perturbations can be used for both detection and removal.

**Weaknesses:**

Comparison with feature-space defenses. While the paper is inspired by Trigger-Activated Changes (TAC), its practical implementation closely resembles feature-space backdoor defenses[a]. However, the paper provides limited comparative analysis with these prior methods. A deeper comparison would strengthen the contribution and clarify the novelty.

Adaptive evaluation. The work does not evaluate the defense under adaptive or defense-aware backdoor attacks. Since the proposed method depends on the assumption that poisoned-class perturbations exhibit smaller L2 norms, an attacker aware of this could manipulate with this regard. Testing against attacks that minimize perturbation norms would provide more substantial evidence of robustness.

Hyperparameters. The defense relies on several dataset-specific hyperparameters, such as the outlier threshold. The paper gives limited guidance on how these parameters generalize across datasets or model architectures. In addition, reproducibility could be improved by reporting computational cost and sensitivity analyses.

[a]Towards Stable Backdoor Purification through Feature Shift Tuning. NeurIPS 2023.

**Questions:**

Compare with feature space defenses, discuss adaptive attacks, discuss the generalization w.r.t hyperparameters

---

> ### Author Response · Authors · 2025-11-28
>
> > **W1. Comparison with feature-space defenses. While the paper is inspired by Trigger-Activated Changes (TAC), its practical implementation closely resembles feature-space backdoor defenses[a]. However, the paper provides limited comparative analysis with these prior methods. A deeper comparison would strengthen the contribution and clarify the novelty.**
>
> We appreciate your suggestion regarding our relationship to feature-space defenses, particularly Feature Shift Tuning (FST) [a]. We believe the reviewer's intuition is reasonable: both FST and our method manipulate latent representations to reduce backdoor influence, and superficially, shifting poisoned features toward clean ones in FST may resemble our use of the reconstructed TAC during fine-tuning. We agree that highlighting these conceptual parallels can help clarify our contribution.
>
> However, despite this high-level similarity, the two approaches differ significantly both in their *qualitative design* and *quantitative behavior*. Conceptually, FST aims to purify latent representations by shifting the poisoned feature distribution toward the clean distribution. In contrast, our method reconstructs TAC as minimal perturbations that directly capture the trigger-induced direction in the latent representation, enabling us to actively suppress backdoor effects. This TAC reconstruction method provides a more explicit understanding of how the backdoor operates internally, which is not provided by FST.
>
> Another important qualitative distinction is that FST does not explicitly identify the poisoned class and therefore modifies the latent space of *all* classes during purification, which may harm clean-model accuracy. In contrast, by detecting the poisoned class through statistically small perturbation norms, our method applies targeted fine-tuning that preserves the accuracy of the clean model almost entirely. This selective nature makes our approach more interpretable and less intrusive than global purification.
>
> Quantitatively, our method also shows more stable behavior across architectures and datasets. As shown in Tables 4 and 5 of the main paper, our defense consistently maintains high clean accuracy while reducing attack success rates across multiple attack types. In comparison, FST exhibits sensitivity to model architectures, particularly VisionTransformers, where feature clusterability is weaker. To further validate this, we conducted additional experiments on ViT in the rebuttal phase, demonstrating that on TinyImageNet our method clearly outperforms FST both in terms of accuracy and attack success rate. These results suggest that our TAC-based suppression is less dependent on architecture-specific feature structures and remains robust in settings where feature-purification approaches become unstable.
>
> In summary, while we acknowledge your insight that both FST and our method operate in latent space and can appear related at a high level, our method differs substantially in both intent and mechanism. FST purifies feature distributions globally, whereas our approach reconstructs TAC to explicitly isolate and suppress trigger-activated directions. This distinction yields more stable empirical performance across datasets, attack types and architectures.

---

> ### Author Response · Authors · 2025-11-28
>
> > **W2. Adaptive evaluation. The work does not evaluate the defense under adaptive or defense-aware backdoor attacks. Since the proposed method depends on the assumption that poisoned-class perturbations exhibit smaller $L^2$ norms, an adversary aware of this could manipulate with this regard. Testing against attacks that minimize perturbation norms would provide more substantial evidence of robustness.**
>
> We appreciate your insightful suggestion regarding adaptive or defense-aware backdoor attacks. We fully agree that evaluating our method under an adversary who explicitly attempts to circumvent the proposed TAC-based detection mechanism is an important aspect of demonstrating robustness.
>
> In particular, as the reviewer pointed out, an adaptive adversary with access to the model training process could deliberately enforce smaller trigger-activated changes (TAC) for the poisoned class. Such an adversary may attempt to directly minimize the $L^2$ norm of the perturbations associated with the poisoned class during model training. A representative adaptive objective can be formulated as follows:
>
> $$ \theta_{\text{bd}}
> = argmin_\theta\,
> \frac{1}{n} \sum_{i=1}^n
> [ \ell(f(x_i;\theta), y_i) + \ell(f(x_i + \delta;\theta), e_t) + \lambda \lVert \phi_{\theta}(x+\delta) - \lambda \phi_{\theta}(x) \rVert_2 ] $$.
> This loss explicitly encourages the adversary to train a model whose latent representations are close between clean data and poisoned data, thereby minimizing the TAC in the latent representation.
>
> To address this defense-aware model, we applied our defense method to the model trained using BadNets with the above objective function.
>
> First, when we tested the poisoned identification method, we confirmed that the contaminated class was correctly identified in all datasets, showing $p_{\text{val}} = 0.000023$ and $N^* = 30$.
>
> Furthermore, the backdoor removal results for CIFAR-10, GTSRB, and TinyImageNet are as shown in below Table:
>
> | Dataset       | No Defense | No Defense | No Defense | Ours | Ours | Ours |
> |---------------|------------|------------|------------|------|-------|-------|
> |               | ACC↑       | ASR↓       | DER↑       | ACC↑ | ASR↓  | DER↑  |
> | CIFAR-10      | 91.90      | 100.00     | -          | 91.35 | 1.09  | 99.18 |
> | GTSRB         | 94.38      | 100.00     | -          | 94.68 | 0.01  | 100.00 |
> | TinyImageNet  | 41.32      | 99.34      | -          | 60.22 | 0.15  | 99.59 |
>
>
> Across all datasets, we find that our defense remains highly effective, reducing ASR to 1.09\%, 0.01\%, and 0.15\%, respectively, while maintaining clean accuracy almost unchanged and increased for GTSRB, TinyImageNet.
> This is because adding the term that minimizes TAC degraded the clean accuracy of the compromised model, and our fine-tuning subsequently restored the accuracy.
>
> These results demonstrate that even when the adversary explicitly attempts to flatten TAC in the latent representation, our defense continues to correctly identify the poisoned class and successfully suppress the backdoor behavior.
>
> We attribute this robustness to two key factors. First, minimizing TAC alone does not eliminate the structural shift that triggers induce in the classifier's decision boundary; our method reconstructs this shift through optimization, rather than relying solely on magnitude-based TAC estimates. Second, our fine-tuning stage explicitly optimizes the model to be invariant to perturbations, which remains effective even when the adversary attempts to minimize the norm.
>
> These experimental results are shown in Appendix F.8.

---

> ### Author Response · Authors · 2025-11-28
>
> > **W3-1. Hyperparameters. The defense relies on several dataset-specific hyperparameters, such as the outlier threshold. The paper gives limited guidance on how these parameters generalize across datasets or model architectures.**
>
>
> We appreciate your suggestion regarding the hyperparameters used in our defense and how they generalize across datasets and architectures. Our method contains two key hyperparameters, and we provide substantial improvements and clarifications to address this point.
>
> First, for the hyperparameter $\alpha$, which was originally used as an outlier threshold on the perturbation norms, we have revised the poisoned-class identification procedure to a more statistically principled formulation. Instead of directly thresholding the smallest perturbation norm, we identify the poisoned class by leveraging the *mode of the minimal-norm class indices* over multiple random subsamplings. This modification introduces three new parameters—$T$, $\eta$, and $\alpha$—that correspond respectively to the number of subsampling rounds, the acceptance ratio in majority voting, and the statistical significance threshold. Importantly, unlike the original $\alpha$, these parameters have clear statistical interpretations and are substantially easier to control. In practice, we found that setting $T_{\text{pci}} = 30$, $\eta = 0.7$, $r=0.4$ and $\alpha = 0.01$ works reliably across all datasets and architectures we tested, indicating that these parameters generalize well and do not require dataset-specific tuning. (See General Response 1 for details.)
>
> Second, for the hyperparameter $\beta$, which balances task accuracy and robustness during fine-tuning, we explored making $\beta$ a **learnable parameter** updated jointly with model weights during optimization. This approach reduces the need for manual tuning, and we observed that it consistently preserves accuracy while lowering the attack success rate. Although automatically learned $\beta$ performs well in most cases, we also found that manually tuned $\beta$ can achieve slightly stronger backdoor removal in some scenarios. Therefore, while our results demonstrate that a partially automated tuning scheme is feasible, devising a fully automatic and uniformly optimal strategy remains an interesting direction for future work. (See General Response 4 for more discussion.)
>
> Finally, these improvements address your concerns by providing more robust, statistically grounded hyperparameter choices, reducing the dependence on dataset-specific tuning, and offering practical strategies for improving reproducibility across architectures and datasets.
>
> > **W3-2. In addition, reproducibility could be improved by reporting computational cost and sensitivity analyses.**
>
> We appreciate your suggestion. Likewise, most of reviewers made similar concerns, so We have provided the response in the General Response 3. Please refer to that part for details.

---

### Author Response · Authors · 2025-11-28
**General Response 1: Modifying the Poisoned Classes Identification Method**

Our current poisoned class identification method relies on a heuristic outlier threshold $\alpha$ on the standardized $L^2$ norms of the class-wise perturbations ${\lVert s_k^* \rVert_2}$.

However, this threshold must inherently be chosen in a data-dependent manner, and the detectability also varies with the strength of the backdoor effect in the latent representation for example, when the poisoning rate is low, the TAC may be too weak for the poisoned class to appear as a clear outlier. As a result, the reliability of this identification approach can be limited.

To address this problem, we replace the current heuristic with a statistical procedure.
This procedure leverages a key empirical property: in compromised models, the poisoned class almost always appears as the class with the smallest perturbation norm ${\lVert s_k^* \rVert_2}$, even when the computation of $s_1, s_2, \cdots, s_C$ is repeated many times.
In contrast, clean models may show small fluctuations across classes, but they never exhibit such a strong and consistent concentration on a single class such as compromised models.

Concretely, we first calculate the latent representations of the entire reference dataset which a defender has in advance. Then, we randomly sample a subset of them at a sampling rate $r$, and obtain $s_1, s_2, \cdots, s_C$ by solving the optimized problem as described in Section 4.1.
This process is repeated $T_{\text{pci}}$ times.
In each process, we record which class attains the minimum norm.
Let $N_k$ be the number of times class $k$ is selected, $N^* = \max_{1 \leq k \leq C} N_k$, and $p_{\max} = N^* / T_{\text{pci}}$.
Then, we introduce a hypothesis test on the maximum selection ratio.
We define a clean model as one in which the maximum selection probability of any class is bounded by $\eta$, i.e., under the null hypothesis.
$$
H_0: p_{\max} \le \eta \quad (\text{clean model}),
$$
while a compromised model violates this bound:
$$
H_1: p_{\max} > \eta \quad (\text{compromised model}).
$$
Since $H_0$ is a composite hypothesis, we evaluate the $p$-value under the *least favorable* clean case, where one class has selection probability exactly $\eta$ and the others are ignored.
In this case, $N^*$ is stochastically dominated by a binomial random variable
$$
X \sim \text{Binomial}(T_{\text{pci}}, \eta),
$$
and we define the $p$-value as

$$p_{\text{val}} = Pr[X >=  N^* ].$$
In other words, we deliberately consider the clean scenario in which a large value of $ N^* $ is most likely to occur, and ask how plausible the observed value is under that scenario. In this way, the $p$-value quantifies how natural it would be, under $H_0$, for a clean model to select the same class as the smallest-norm class as many times as we observed.
We then declare the model compromised if $p_{\text{val}} < \alpha$, and identify the poisoned class as the one with the largest $N_k$. Otherwise, we regard the observed bias as consistent with a clean model.
**If the poisoned class $t$ is identified, we compute the perturbation $s_t^*$ using the entire reference dataset, so the results using our fine-tuning method can remain the current results.**

This modification addresses the reviewers' concerns in two ways.
(i) The threshold $\alpha$ is now a **standard significance level** controlling the error (false detection on clean models), rather than a dataset-specific z-score cutoff.
The dataset dependence is concentrated in $\eta$, which has a clear interpretation as the maximum allowable bias in clean models; it can be chosen from the empirical upper bound of $p_{\max}$ over a few clean models with a safety margin, yielding a data-adaptive yet principled detector.
(ii) By aggregating evidence over many trials and focusing on the extreme concentration of the minimum-norm class, the method becomes more sensitive to weak triggers and low poisoning rates: even when the per-trial norm gap is small, the poisoned class tends to win slightly but consistently, leading to a statistically significant bias in $N_k$.

---

> ### Author Response · Authors · 2025-11-28
>
> We further provide empirical evidence that the proposed hypothesis-testing procedure reliably separates clean and compromised models across various attack types. Using $T_{\text{pci}} = 30$, significance level $\alpha = 0.01$, sampling rate $r=0.4$, and clean-model upper-bound parameter $\eta = 0.7$, we obtain the following results for CIFAR-10, GTSRB, and ImageNet on ResNet-18:
>
> | Attack Method | CIFAR-10 | CIFAR-10 | CIFAR-10 | GTSRB | GTSRB | GTSRB | TinyImageNet | TinyImageNet | TinyImageNet |
> | -------- | -------- | -------- | -------- |  -------- | -------- | -------- |  -------- | -------- | -------- |
> |  | $p_{val}$  | $N^*$ | Poisoned Class | $p_{val}$  | $N^*$ | Poisoned Class | $p_{val}$  | $N^*$ | Poisoned Class |
> | Clean   | 0.998 | 14  | - | 0.589 | 21 | - | 0.841 | 19 | -
> | BadNets | 0.0000226  | 30 | 1 |  0.0000226  | 30 | 1 |  0.0000226  | 30 | 1 |
> | Trojan  | 0.0000226  | 30 | 1 |  0.0000226  | 30 | 1 |  0.0000226  | 30 | 1 |
> | Blend   | 0.0000226  | 30 | 1 |  0.0000226  | 30 | 1 |  0.0000226  | 30 | 1 |
> | WaNet   | 0.0000226  | 30 | 1 |  0.0000226  | 30 | 1 |  0.0000226  | 30 | 1 |
> | IAB     | 0.0000226  | 30 | 1 |  0.0000226  | 30 | 1 |  0.0000226  | 30 | 1 |
> | Lira    | 0.0000226  | 30 | 1 |  0.0000226  | 30 | 1 |  0.0000226  | 30 | 1 |
>
> The clean model exhibits only $N^* =20$ for CIFAR-10, resulting in an extremely large $p_{\text{val}}$ and thus is not rejected under $H_0$. In contrast, differing trigger types and injection mechanisms show perfect concentration ($N^* = 30$) in all datasets, yielding highly significant $p$-values ($<10^{-3}$). These results confirm that the proposed method consistently identifies the poisoned class and cleanly separates compromised models from clean ones.
>
> **The revised proposed method has been reflected in all relevant sections of the paper, including the abstract, contributions, overview, proposed method, experimental results, conclusion, and appendix.**

---

### Author Response · Authors · 2025-11-28
**General Response 2: Large-scale Model and Dataset**

We appreciate the suggestions from most of reviewers regarding the importance of evaluating the proposed method on large-scale architectures (e.g., Vision-Transformers) and large datasets (e.g., ImageNet-1K). We fully agree that demonstrating robustness beyond ResNet-based models is essential for verifying the generality and practicality of our approach.

To address this concern, we conducted additional experiments using **ViT-B/32** on **TinyImageNet** and **ImageNet-1K**. In all attacks, we fine-tune the pre-trained models starting from publicly available pretrained ViT-B/32 checkpoints. We also included comparisons with the defenses, **FST**, **SAU**, and **FT-SAM**.

The results for TinyImageNet are presented in the table below:

| Attack | No Defense | No Defense | FT-SAM | FT-SAM | SAU | SAU | FST | FST | Ours | Ours |
|--------|------|------|------|------|------|------|------|------|------|------|
|        | ACC↑ | ASR↓ | ACC↑ | ASR↓ | ACC↑ | ASR↓ | ACC↑ | ASR↓ | ACC↑ | ASR↓ |
| BadNets | 62.76 | 100.00 | 60.70 | 100.00 | 53.35 | 6.16 | 39.13 | 2.16 | 59.99 | 7.32 |
| Trojan  | 61.87 | 100.00 | 58.94 | 100.00 | 52.72 | 0.01 | 35.11 | 38.74 | 59.33 | 0.48 |
| Blend   | 62.37 |  99.99 | 60.04 | 99.91 | 53.92 | 0.03 | 35.54 | 59.87 | 58.22 | 0.00 |
| Average | 62.33 |  99.99 | 59.89 | 99.97 | 53.33 | 2.07 | 36.59 | 33.59 | 59.18 | 2.60 |



We also show the result for ImageNet-1K in the table below:

| Attack | No Defense | No Defense | FT-SAM | FT-SAM | SAU | SAU | FST | FST | Ours | Ours |
|--------|------|------|------|------|------|------|------|------|------|------|
|        | ACC↑ | ASR↓ | ACC↑ | ASR↓ | ACC↑ | ASR↓ | ACC↑ | ASR↓ | ACC↑ | ASR↓ |
| BadNets | 70.52 | 100.00 | 69.01 | 100.00 | 51.16 | 0.28 | 58.54 | 0.01 | 59.54 | 0.00 |
| Trojan  | 71.33 | 100.00 | 70.15 | 99.95 | 51.40 | 0.01 | 59.83 | 0.00 | 60.52 | 0.00 |
| Blend   | 71.48 | 99.98  | 70.11 | 99.57 | 52.36 | 0.01 | 59.38 | 0.00 | 60.42 | 0.01 |
| Average | 71.11 | 99.99  | 69.76 | 99.84 | 51.64 | 0.10 | 59.25 | 0.00 | 60.16 | 0.00 |




For TinyImageNet, our method consistently achieves a favorable balance between ACC and ASR. For instance, while SAU and FST achieve low ASR for certain attacks, their ACCs degrade substantially, dropping to 53\% and 36\% on average, respectively. In contrast, our method maintains a high ACC of 59.18\% comparable to FT-SAM, while achieving a much lower ASR of 2.60\%, far outperforming FT-SAM whose ASR remains above 99\%. These findings indicate that, even in a large-scale model setting, the proposed method retains its effectiveness.

We further extended our evaluation to ImageNet-1K, which is an order of magnitude more challenging, involving 1000 classes and high-dimensional features characteristic of large pretrained models. Remarkably, our method continues to perform strongly. Across BadNets, Trojan, and Blend attacks, our defense reduces ASR to an average of 0.003\%, matching or improving upon SAU and FST, both of which also achieve low ASR in this setting. Crucially, however, our method maintains a ACC of 60.16\%, which is dramatically higher than SAU (51.64\%) and substantially higher than FST (59.25\%). Meanwhile, FT-SAM remains ineffective on ViT-B/32, with ASR exceeding 99\% across all three attacks. These ImageNet-1K results demonstrate that our approach scales gracefully even in extremely large-scale dataset settings and remains robust.

Finally, these experiments provide strong evidence that the proposed defense generalizes beyond ResNet-based models and small datasets. Its effectiveness on ViT-B/32, combined with stable performance on both TinyImageNet and ImageNet-1K, confirms that the TAC reconstruction and fine-tuning mechanism does not rely on CNN-specific structures and remains robust even when applied to high-capacity architectures and large, complex datasets. This further reinforces the practicality and universality of our defense in real-world scenarios where large models and large-scale data are standard.


These experimental results are provided in Appendix F.1.

---

### Author Response · Authors · 2025-11-28
**General Response 3: Time-consuming of Our Proposed Method**

Most of reviewers have pointed out that solving one quadratic program (QP) per class during TAC reconstruction may raise concerns about computational cost and practicality. We fully agree that runtime efficiency is an important aspect for a deployable defense method, and here we provide a detailed clarification based on both computational complexity analysis and empirical measurements.


We begin by describing the computational complexity of the QP we solve for each class, as defined in equation 5 of the main paper:

$$\lambda^* = \arg\max_{\lambda}
\left( \lambda^\top m - \frac{1}{2} \lVert V^\top \lambda \rVert_2^2 \right).
$$

The dominant term in evaluating the dual objective is the computation of $\| V^\top \lambda \|^2_2 = \lambda^\top (VV^\top) \lambda = (V^\top \lambda)^\top (V^\top \lambda)$, where $V \in \mathbb{R}^{(C-1)\times d_{\text{emb}}}$.
The matrix vector product $V^\top \lambda$ requires $O(d_{\text{emb}} C)$ operations, and the linear term $\lambda^\top m$ adds $O(C)$. Thus, each iteration of the convex solver costs:$O(d_{\text{emb}} C)$.
If the solver performs $T_\text{solver}$ iterations, the per-class cost becomes:$O(T_\text{solver} d_{\text{emb}} C)$.

Importantly, this complexity depends only on the dimension $d_{\text{emb}}$ and the number of classes $C$; it does not depend on the size of the dataset or network depth. Moreover, the dimension is typically modest (e.g., 512 or 2048 for ResNet-based models), which keeps the optimization efficient even for large-scale classification problems.


To supplement the complexity analysis, we also report actual time measurements for computing a minimal perturbation for per class in the following Table. Note that only the CPU, not the GPU, was used when computing the QP.


| Dataset      | ResNet-18 ($d_{\text{emb}}=512$) | ResNet-50 ($d_{\text{emb}}=2048$) | ViT-B/32 ($d_{\text{emb}}=512$) |
| ------------ | ---------------------- | ----------------------- | --------------------- |
| CIFAR-10     | 0.015388 ± 0.000639    | 0.032364 ± 0.003099     | 0.015485 ± 0.000380   |
| GTSRB        | 0.032178 ± 0.004127    | 0.104500 ± 0.002913     | 0.033923 ± 0.003504   |
| TinyImageNet | 0.141203 ± 0.002386    | 0.474484 ± 0.021590     | 0.142614 ± 0.004958   |
| ImageNet-1K  | -                  | -     | 1.170394 ± 0.018746   |


As a result, we confirmed that the solving time for a single class is very short and practical. Even for ImageNet-1K, which contains 1,000 classes, the computation runs fast---approximately 1.1 seconds for $d_{\text{emb}} = 512$.

**Our poisoned class identification method requires additional computations: specifically, $C$ runs for the number of classes and $T_{\text{pci}}$ runs for the statistically necessary number of trials. A straightforward implementation would require $C T_{\text{pci}}$ additional computations, but since each computation is completely independent, they can be executed in parallel.**

As a result, both the complexity analysis and the runtime measurements show that the proposed TAC reconstruction method is computationally lightweight and practical, even when scaling to larger numbers of classes or higher-dimensional embeddings.

These experimental results are provided in Appendix E.

---

### Author Response · Authors · 2025-11-28
**General Response 4: Automatically Tuning Hyperparameter β**

We thank the reviewers (norE and SXdY) for raising the concern regarding the tuning or generalization of the hyperparameter $\beta$, which balances backdoor removal and clean accuracy during fine-tuning. We agree that reducing the reliance on manual tuning is important for improving the usability and robustness of our proposed method.

In our additional experiments, we found that $\beta$ can be treated as a learnable scalar parameter and optimized jointly with the model parameters during the fine-tuning stage. Specifically, we initialize $\beta$ with a reasonable default value and update it through gradient-based optimization along with the model weights. In the table below, we show the experimental results compared to the baseline results (Manual) fixed at $\beta=0.5$ for CIFAR-10 on ResNet-18.
We use the initial $\beta=\log (1+e^{u})$ with $u=3.0$, so that a softplus function is applied to prevent $\beta$ from taking negative values.
Here, we directly update $u$, which serves as an input to the softplus function.


| Attack Method | No Defense | No Defense | Ours (Manual) | Ours (Manual) | Ours (Auto) | Ours (Auto) |
| ------------- | ---------- | ---------- | ------------- | ------------- | ----------- | ----------- |
|               | ACC↑       | ASR↓       | ACC↑          | ASR↓          | ACC↑        | ASR↓        |
| BadNets       | 93.81      | 100.00     | 92.03         | 10.88         | 91.28       | 1.34        |
| Trojan        | 94.00      | 100.00     | 92.01         | 0.98          | 91.41       | 1.29        |
| Blend         | 93.29      | 99.91      | 91.84         | 1.69          | 91.13       | 0.87        |
| WaNet         | 93.41      | 99.59      | 92.39         | 0.52          | 91.88       | 0.69        |
| IAB           | 93.57      | 98.81      | 92.37         | 0.38          | 91.83       | 0.46        |
| Lira          | 94.29      | 99.98      | 92.80         | 0.11          | 91.82       | 0.19        |



Although the initial value of $\beta$ or $u$ must be specified, we observed that $\beta$ rapidly converges to a stable range regardless of its initialization and that the resulting performance is comparable to that obtained with manually tuned $\beta$.
Here, we confirmed that each final value of $\beta$ does not collapse to 0 and remains meaningful.
It is possible that, during training, the loss term multiplied by $\beta$ became quite small and eventually received almost no updates.
In summary, our results of the simple automatic tuning indicate that a large portion of the tuning effort can be automated without requiring dataset- or model-specific heuristics.

These experimental results including GTSRB and TinyImageNet are provided in Appendix F.6.

---

### Author Response · Authors · 2025-11-28

Dear reviewers,

Thank you very much for your thoughtful and constructive feedback. We have revised the manuscript accordingly, and all modifications are highlighted in red. Your comments were invaluable in improving the clarity and overall quality of the paper.

We also apologize for the delay in our response; we took additional time to carefully address each point in depth to provide the most accurate and meaningful clarifications.

We sincerely appreciate your time and effort in reviewing our work.

Best regards,

Authors

---

### Meta-Review · Area_Chair_rEd2 · 2025-12-24

**Summary:**

Reviewers primarily questioned the generality and practicality of the proposed defense. Early concerns focused on scalability, as the original submission evaluated mainly small CNNs and datasets, leaving uncertainty about computational cost and effectiveness on large architectures and datasets. Reviewers also noted that the method is designed for single-target backdoor attacks and does not naturally handle multi-target or all-to-all attacks, limiting its coverage of more general threat models.

Additional concerns involved robustness and assumptions, including whether the reliance on perturbation-norm disparities for poisoned-class identification would hold under adaptive or defense-aware attacks. Reviewers further pointed out potential hyperparameter sensitivity, questioning whether the method requires dataset- or model-specific tuning and how reproducible it is in practice.

Finally, some reviewers asked whether the method offers clear advantages over strong existing baselines (e.g., SAU, FST, FT-SAM), especially in settings where those methods already achieve low attack success rates.

In the rebuttal, the authors addressed several of these points by adding large-scale experiments (ViT, ImageNet-1K), providing runtime analysis, and introducing a more principled statistical identification procedure. Nonetheless, limitations remain, particularly regarding all-to-all attacks and the reliance on structural assumptions about backdoor behavior, which reviewers considered important open issues.

**Reviewer Concerns:**

Concerns addressed by the rebuttal.
The rebuttal and revised manuscript convincingly addressed several major reviewer concerns. First, the authors substantially improved the scalability and generality of the evaluation by adding experiments on ViT-B/32 and ImageNet-1K, demonstrating that the method remains effective beyond small CNNs and mid-scale datasets. Second, concerns regarding computational overhead were largely resolved through detailed complexity analysis and empirical runtime measurements, showing that the per-class convex optimization is practical and parallelizable. Third, issues related to hyperparameter sensitivity were mitigated by replacing heuristic thresholds with a statistically grounded poisoned-class identification test and by introducing an automatic tuning strategy for the fine-tuning balance parameter. Finally, reviewers’ requests for adaptive or defense-aware attack evaluation were addressed with additional experiments showing that the method remains effective even when attackers attempt to suppress TAC magnitude.

Concerns still outstanding.
Some limitations remain unresolved. Most notably, the method still cannot reliably identify poisoned classes under multi-target or all-to-all backdoor attacks; while TAC reconstruction and fine-tuning can be extended assuming the attack is known, detection in this setting remains an open problem. In addition, although comparisons with strong baselines were strengthened on large-scale settings, the performance advantage over existing methods is less pronounced on small CNNs, where several baselines already perform competitively. These issues reflect inherent limitations of the current formulation rather than missing experiments, and were appropriately acknowledged by the authors as directions for future work.

**Reviewer Scores:**

Reviewer norE:
Originally rated the paper negatively, with concerns about novelty relative to feature-space defenses, hyperparameter sensitivity, scalability, and lack of adaptive-attack evaluation. Given that the rebuttal added large-scale ViT/ImageNet experiments, runtime analysis, adaptive-attack results, and a more principled statistical identification method, this reviewer would likely increase the score modestly, but may still remain below or around the borderline, due to persistent concerns about conceptual overlap with prior feature-space methods and limited gains on small CNNs.

Reviewer vvLe:
Initially marginally positive but raised concerns about limited architectures, datasets, and assumptions of a single poisoned class. Since the rebuttal directly addressed scalability with ViT/ImageNet results and clarified the limitation on all-to-all attacks, this reviewer would likely increase the score slightly, moving from marginally above threshold to more confidently positive, while still noting the unresolved multi-target identification issue.

Reviewer sDFY:
Provided one of the most favorable assessments, emphasizing the novelty and mathematical grounding of TAC reconstruction, but requested scalability evidence and runtime analysis. As these points were comprehensively addressed in the rebuttal, this reviewer would likely maintain or slightly increase an already positive score, remaining supportive of acceptance.

Reviewer SXdY:
Originally marginally negative to borderline, with concerns about scalability, hyperparameter tuning, multi-target attacks, and low poisoning rates. The rebuttal resolved many practical concerns (runtime, adaptive tuning, large-scale experiments), which would likely lead this reviewer to raise the score to around the acceptance threshold, though reservations about all-to-all attack detection would probably persist.

---

### Decision · Program_Chairs · 2026-01-26

Reject